# Gaming and Cooperation in Federated Learning: What Can Happen and How to Monitor It

**Dongseok Kim**\*                                     *jkds5920@gachon.ac.kr*
*Department of Computer Engineering*
*Gachon University*

**Hyoungsun Choi**\*                                  *hschoi@gachon.ac.kr*
*Department of Computer Engineering*
*Gachon University*

**Mohamed Jismy Aashik Rasool**\*                 *aashikrasool@gachon.ac.kr*
*Department of Computer Engineering*
*Gachon University*

**Gisung Oh**†                                         *eustia@gachon.ac.kr*
*Department of Computer Engineering*
*Gachon University*

**Reviewed on OpenReview:** *https://openreview.net/forum?id=Ck3q5YdWIv*

## Abstract

The success of federated learning (FL) ultimately depends on how strategic participants behave under partial observability, yet most formulations still treat FL as a static optimization problem. We instead view FL deployments as governed strategic systems and develop an analytical framework that separates welfare-improving behavior from metric gaming. Within this framework, we introduce indices that quantify manipulability, the price of gaming, and the price of cooperation, and we use them to study how rules, information disclosure, evaluation metrics, and aggregator-switching policies reshape incentives and cooperation patterns. We derive threshold conditions for deterring harmful gaming while preserving benign cooperation, and for triggering auto-switch rules when early-warning indicators become critical. Building on these results, we construct a design toolkit including a governance checklist and a simple audit-budget allocation algorithm with a provable performance guarantee. Simulations across diverse stylized environments and a federated learning case study consistently match the qualitative and quantitative patterns predicted by our framework. Taken together, our results provide design principles and operational guidelines for reducing metric gaming while sustaining stable, high-welfare cooperation in FL platforms.

## 1 Introduction

### 1.1 Motivation

Federated learning (FL) enables multiple organizations to train a shared model without moving data, and adoption has been growing through consortia and platform-based collaborations. As cross-organizational data collaboration expands, keeping data local while jointly training models has become a leading alternative to traditional data pooling, especially when privacy, confidentiality, and regulatory constraints rule out centralization. In this setting, FL naturally supports emerging data and AI services in which participants

---

\*These authors contributed equally.
†Corresponding author.

retain control over their local data while contributing to shared intelligence that is monetized or governed through metrics, contracts, and service-level guarantees.

At the same time, the surrounding governance has not matured at the same pace. Many organizations are still in early stages of formalizing procedures for AI-enabled services, clarifying accountability, and embedding risk mitigation into day-to-day operations. In FL, these gaps are particularly consequential: joint outcomes depend on the actions of multiple parties whose internal processes are opaque to one another, yet whose rewards and reputation are coupled through shared metrics. When rewards, rankings, or access rights depend on these metrics, participants face incentives not only to improve genuine performance but also to target the metrics themselves, potentially drifting toward high-metric, low-welfare regimes.

This tension is amplified by the combination of privacy and limited observability. Privacy-enhancing technologies such as FL, differential privacy, and secure computation restrict what can be inspected or audited, and information design around metrics and scores further shapes what participants see and react to. Strong protections are desirable, but they also reduce the visibility of individual behavior and may unintentionally make harmful manipulation harder to detect. Designing FL systems therefore requires more than choosing an optimizer or aggregation rule: it requires a coordinated view of evaluation, information disclosure, rewards, audits, and participation, and of how these choices jointly shape incentives, cooperation, and stability.

This paper aims to provide such a view. We treat FL as a governed strategic system and develop a three-layer framework that (i) quantifies how design choices create room for metric gaming and cooperation, (ii) links these quantities to participation dynamics and tipping points, and (iii) maps them to practical levers such as mixed public–private evaluation, audit allocation, sanction calibration, and auto-switch rules. Our goal is not to propose yet another FL algorithm, but to offer a compact language and toolkit that helps designers reason about metric gaming, cooperation, and governance in federated environments.

## 1.2 Contributions

We view federated learning not merely as a distributed optimization problem but as a *governed strategic system* in which evaluation rules, information disclosure, reward and sanction mechanisms, and audit capacity jointly shape participants' incentives. Our contributions are to provide a formal language for this system, to connect that language to simple indices and thresholds that can be estimated from data, and to distill resulting design principles into an actionable toolkit.

- **Strategic formalization of federated learning.** We formalize a generic *Eval–Info–Reward–Audit* architecture for federated learning platforms (Section 3), specifying welfare, public and private metrics, participation choices, and cooperative actions (e.g., coalition formation, data sharing) within a single game-theoretic environment. This provides a common backbone on which existing robust aggregation, incentive, and privacy mechanisms can be placed.

- **Indices for manipulability, gaming, and cooperation.** On top of this backbone, we introduce three indices that summarize how a design policy $\pi$ trades off metric performance and welfare: a manipulability index $M(\pi)$ that captures the best achievable metric gain per unit welfare loss under unilateral deviations, a Price of Gaming $\mathrm{PoG}(\pi)$ that measures welfare loss when a fraction of clients adopt gaming behaviors, and a Price of Cooperation $\mathrm{PoC}$ that quantifies the net welfare effect of coalitions (Section 4). We establish basic properties of these indices, show how they interact with simple penalty schemes, and use them to define design-time thresholds $(\alpha_{\min}, \alpha_{\mathrm{benign}})$ that separate under-enforcement, harmful gaming regimes, and over-enforcement that discourages benign cooperation.

- **Participation dynamics, resilience, and tipping thresholds.** We couple the metric layer to a stylized participation dynamics model, defining an aggregate participation map $F(x; \pi)$ and a resilience indicator $R(\pi)$ that summarize how participation responds to policy changes (Section 5). Under this model, we characterize tipping points and domino-style exits, and show how the indices above bound the regions in which small changes in sanction strength or public-metric weight can trigger large shifts in participation. This yields interpretable thresholds and heuristic rules for maintaining participation stability while suppressing high-PoG equilibria.

- **Design toolkit for evaluation, audits, and governance.** Building on these indices and dynamics, we propose a set of design patterns for federated platforms (Section 6), including mixed public/private evaluation schemes, audit-budget allocation rules with $(1-1/e)$ approximation guarantees for detecting high-manipulability clients, and a governance checklist that links observable diagnostics (e.g., gaming incident rates, participation responses, coalition structures) to concrete levers (metric choice, disclosure policy, sanction strength, and audit allocation).

- **Empirical illustrations in stylized and federated settings.** Finally, we instantiate the framework in a stylized simulator and a federated Fashion-MNIST experiment (Section 7), showing how high-metric/low-welfare equilibria, participation tipping, and benign versus harmful cooperation patterns arise under different design policies. We demonstrate that our indices can be approximated using simple scenario-based experiments and log-based diagnostics, and that the proposed toolkit can detect and mitigate high-PoG regimes without collapsing participation.

## 2 Related Work

### 2.1 FL Attacks, Defenses, and Robust Aggregation

Early work on robustness in federated and distributed learning asked whether a small number of malicious or corrupted updates can derail training, leading to robust aggregation rules such as Krum, coordinate-wise median and trimmed mean, Bulyan, and Robust Federated Aggregation (RFA) (Blanchard et al., 2017; Yin et al., 2018; Guerraoui et al., 2018; Pillutla et al., 2022), along with convergence analyses under Byzantine noise and coding-based defenses (Alistarh et al., 2018; Bernstein et al., 2018; Chen et al., 2018). Subsequent studies implemented these mechanisms under realistic constraints and exposed their limits via model-poisoning, backdoor, and tail-group attacks, and proposed collusion- and Sybil-aware defenses such as FoolsGold and FLTrust (Damaskinos et al., 2019; Xie et al., 2020; Baruch et al., 2019; Fang et al., 2020; Bagdasaryan et al., 2020; Wang et al., 2020; Fung et al., 2018; Cao et al., 2020). Our work does not add another aggregation or poisoning defense; instead, we ask how the choice of rules, metrics, and audits shapes incentives for manipulation and cooperation, and how these incentives affect aggregate performance and participation even when the underlying optimization dynamics are well behaved.

### 2.2 Incentive Design and Differential Client Contribution

Differential contribution and reward design often start from data valuation: Shapley-value-based methods, influence functions, and learned value estimators seek fair allocations by tracing marginal contributions of data or clients (Ghorbani & Zou, 2019; Jia et al., 2019; Koh & Liang, 2017; Yoon et al., 2020; Liu et al., 2022b; Chen et al., 2023; 2024; Tastan et al., 2024). Direct incentive schemes use reputation, contracts, auctions, and blockchain-based mechanisms to induce participation and effort, while parallel work folds contribution weighting into learning and aggregation to target fairness, robustness, or resource-aware client selection (Kang et al., 2019; Zhang et al., 2021a; Tang et al., 2024; Zhang et al., 2021b; Liu et al., 2020; Mohri et al., 2019; Li et al., 2019; Lai et al., 2021; Nishio & Yonetani, 2019; Lin et al., 2023; Kim et al., 2024; Ouyang & Kuang, 2025). These approaches typically treat incentives as reward-allocation functions on fixed metrics or as modified aggregation rules; in contrast, we focus on the strategic environment induced by evaluation and audit design itself and introduce indices that quantify when such designs incentivize gaming versus genuine improvement.

### 2.3 Game Theory of Federated Learning and Participation Dynamics

Game-theoretic treatments of FL have emphasized coalition formation, stability, and free-riding, using hedonic games, coalition models, and cooperative-game valuations to study gaps between individually stable and globally optimal outcomes (Donahue & Kleinberg, 2021a;b; Hasan, 2021; Nagalapatti & Narayanam, 2021). Repeated-game and leader–follower analyses show how punishment strategies, contract-theoretic mechanisms, and Stackelberg formulations can deter free-riding and align server–client incentives under private information, including collusion- and Sybil-aware variants (Zhang et al., 2022; Sagduyu, 2022; Luo et al.,

2023; Sarikaya & Ercetin, 2019; Hu & Gong, 2020; Le et al., 2021; Ding et al., 2020; Liu et al., 2022a; Huang et al., 2024; Byrd et al., 2022; Xiong et al., 2024). Our work shares the strategic perspective but shifts focus from specific mechanism equilibria to a metric-based language and threshold results for participation stability, tipping points, and domino exits under generic rule sets.

## 2.4 Metric Gaming and the Goodhart Phenomenon

Metric gaming is often summarized by Goodhart's observation that once a measure becomes a target, it ceases to be a good measure. Prior work has categorized mechanisms behind this phenomenon and documented failure modes such as reward hacking, non-scalable oversight, and goal misgeneralization (Manheim & Garrabrant, 2018; Amodei et al., 2016; Skalse et al., 2022; Everitt et al., 2021; Di Langosco et al., 2022). When metrics drive decisions and thereby change the data, the problem becomes strategic: models of strategic classification, Stackelberg interactions, and performative prediction analyze how agents respond to public classifiers and how repeated retraining interacts with distributional shifts (Hardt et al., 2016a; Brückner & Scheffer, 2011; Perdomo et al., 2020; Mendler-Dünner et al., 2020; Miller et al., 2020). We build on these insights but specialize them to FL, focusing on how evaluation metrics, disclosure policies, and audits induce client-level incentives for metric targeting and introducing explicit indices and thresholds that distinguish welfare-improving behavior from gaming.

## 2.5 Design of Evaluation Information, Audits, and Sanctions

Information design around metrics and scores has been proposed as a primary tool against overfitting and gaming, via staircase leaderboards, reusable holdouts, and adaptive-data-analysis bounds for safe repeated evaluation (Blum & Hardt, 2015; Dwork et al., 2015b;c;a; Bassily et al., 2016). From the audit side, work on fairness, documentation, behavior-based testing, and distribution-shift benchmarks has developed procedures and artifacts that surface vulnerabilities beyond single scalar metrics (Kearns et al., 2018; Hardt et al., 2016b; Kim et al., 2019; Agarwal et al., 2018; Kusner et al., 2017; Mitchell et al., 2019; Gebru et al., 2021; Ribeiro et al., 2020; Kiela et al., 2021; Koh et al., 2021; Croce et al., 2020). Privacy and memorization audits—including membership inference, memorization measures, data extraction, and backdoor detection—further highlight the role of audits and sanctions in deployed systems (Shokri et al., 2017; Carlini et al., 2019; 2021; Tran et al., 2018; Wang et al., 2019; Rabanser et al., 2019; Dressel & Farid, 2018). These strands provide building blocks for evaluation and auditing but are usually studied separately from incentives and participation; our contribution is to integrate evaluation information design, audit allocation, and sanction triggers into a single FL framework with indices and thresholds linked to gaming incentives and cooperation stability.

## 2.6 Trade-offs Between Privacy and Incentives

Privacy in FL is commonly enforced through client-level differential privacy and secure aggregation, often combined with distillation or selective-sharing protocols to limit exposure of sensitive data (Abadi et al., 2016; McMahan et al., 2018; Geyer et al., 2017; Bonawitz et al., 2017; Erlingsson et al., 2019; Papernot et al., 2017; 2018; Shokri & Shmatikov, 2015; Gilad-Bachrach et al., 2016). Refined attack models, including gradient inversion, attribute inference, backdoor and membership attacks, have clarified the limits of these protections and the tension between strong privacy, visibility of malicious behavior, and group-level performance and fairness (Melis et al., 2019; Zhu et al., 2019; Bagdasaryan et al., 2020; Wang et al., 2020; Bagdasaryan et al., 2019; Jagielski et al., 2019; Tramer & Boneh, 2020). Recent work on proofs of correct training and bidirectional verification protocols aims to restore some auditability under privacy constraints (Jia et al., 2021; Zhang & Yu, 2022), but typically remains orthogonal to incentive design. Our framework explicitly foregrounds the trade-offs between privacy, audit signals, and incentives, and proposes design patterns—such as randomized evaluation, limited disclosure, and connectivity-based alarms—that help maintain incentive alignment and sanctionability under realistic privacy and legal constraints. While prior work typically focuses on individual mechanisms, attacks, or defenses, our contribution is to provide a platform-level framework that jointly links metric gaming, welfare loss, participation dynamics, and governance levers through explicit indices and design thresholds.

# 3 Federated Learning as a Strategic System: Setup and Notation

In this section we formalize federated learning (FL) as a strategic system. The goal is not a fully realistic economic model, but a minimal structure that (i) makes precise what clients can do strategically, (ii) separates *true welfare* from *observable metrics*, and (iii) allows design choices in evaluation, information, rewards, and audits to be treated as policy variables. Our aim is to isolate the *FL-specific* sources of strategic tension: (i) repeated, round-based interaction; (ii) aggregation, where each reported update affects a shared global model; and (iii) partial observability, where rewards and sanctions are functions of evaluation pipelines rather than direct welfare. Concretely, we treat evaluation, information disclosure, rewards, and audits as explicit policy components $\pi = (\mathsf{Eval}, \mathsf{Info}, \mathsf{Reward}, \mathsf{Audit})$ that are coupled to the learning protocol through $\mathsf{Agg}_t$ and the induced metric process $\{M_t\}$.

## 3.1 Federated environment and timing

We consider a cross-silo FL environment with a finite set of clients

$$\mathcal{I} = \{1, \ldots, n\}$$

and a coordinating server. Time is discrete,

$$t = 1, 2, \ldots, T,$$

where $T$ may be finite or infinite. At each round $t$, the server maintains a global model

$$\theta_t \in \mathbb{R}^d,$$

and each client $i$ holds a local dataset $D_i$ drawn from an unknown distribution $\mathcal{P}_i$. We write $\mathcal{P} = \{\mathcal{P}_i\}_{i \in \mathcal{I}}$ for the collection of client distributions and $P^\star$ for the target population distribution of interest.

A generic round proceeds as follows:

1. **Broadcast:** the server broadcasts $\theta_t$ to eligible clients.

2. **Participation and action:** each client chooses participation $p_{i,t} \in \{0, 1\}$ and, if participating, a local training/reporting behavior.

3. **Local computation:** participating client $i$ computes an internal update $u_{i,t}^{\mathrm{int}} \in \mathbb{R}^d$ from $(D_i, \theta_t)$.

4. **Reporting:** client $i$ reports an update $u_{i,t} \in \mathbb{R}^d$, which may equal $u_{i,t}^{\mathrm{int}}$ (honest), be perturbed (privacy/obfuscation), or be strategically chosen.

5. **Aggregation and evaluation:** the server aggregates updates from $\mathcal{I}_t = \{i : p_{i,t} = 1\}$ via $\mathsf{Agg}_t$ and evaluates via $\mathsf{Eval}_t$ to produce observable signals.

6. **Rewards, audits, sanctions:** the server assigns rewards, selects audits, and applies sanctions based on observable signals and history.

This timing highlights the key FL dependence: the designer does not observe client-side behavior directly, but only the outputs of $\mathsf{Eval}_t$ applied to aggregated models and challenge data. Thus, incentives operate through a mediated channel (scores, disclosures, and audits) rather than through welfare itself, and strategic behavior targets whatever components of $O_t$ are disclosed and rewarded.

The tuple

$$\mathcal{E} = \left( \mathcal{I}, \mathcal{P}, P^\star, \{D_i\}_{i \in \mathcal{I}}, \{\theta_t\}_{t \geq 1}, \{\mathsf{Agg}_t\}_{t \geq 1} \right)$$

specifies the algorithmic FL environment. Strategic behavior is determined by how actions, metrics, rewards, and audits are defined on top of $\mathcal{E}$.

### 3.2 Actions, strategies, metrics, and welfare

**Actions and strategy space.** At each round $t$, client $i$ chooses an action

$$a_{i,t} \in \mathcal{A}_i,$$

where $\mathcal{A}_i$ is a feasible action set. We model $a_{i,t}$ as encoding: (i) participation $p_{i,t} \in \{0, 1\}$, (ii) local training effort/policy that determines $u_{i,t}^{\text{int}}$, and (iii) a reporting/manipulation policy that maps $(\theta_t, u_{i,t}^{\text{int}}, D_i, \text{history})$ to a reported update $u_{i,t}$ and auxiliary reports.

Let $h_t$ denote the public history up to and including round $t$ (e.g., disclosed scores, sanctions, aggregate statistics). A *(behavioral) strategy* for client $i$ is a mapping

$$\sigma_i : h_{t-1} \mapsto a_{i,t}.$$

Let $\Sigma_i$ denote the set of admissible strategies for client $i$, and define the ambient strategy space

$$\Sigma := \prod_{i \in \mathcal{I}} \Sigma_i.$$

**Reference classes (threat models).** Some of our indices restrict attention to a subset of feasible deviations, such as bounded gaming intensity, bounded coalition size, or actions compatible with a given protocol.

**Definition 3.1** (Reference class). A *reference class* is a subset $\Sigma^{\text{ref}} \subseteq \Sigma$ specifying the deviations (or strategy profiles) deemed feasible in the deployment under study. Indices defined via worst-case deviations are interpreted as restricted to $\Sigma^{\text{ref}}$.

**True outcomes and welfare.** Given a strategy profile $\sigma \in \Sigma$ and environment $\mathcal{E}$, the induced model trajectory $\{\theta_t\}$ yields time-dependent *true welfare* on $P^\star$:

$$W_t(\sigma) := W(\theta_t; P^\star),$$

where $W$ is a task-dependent welfare functional. We summarize long-run welfare by an aggregate functional

$$W(\sigma) := \Phi(\{W_t(\sigma)\}_{t \geq 1}),$$

such as a steady-state value or discounted average.

**Metrics and proxy performance.** Rewards, sanctions, and policy changes depend on *metrics* computed from finite evaluation sets and partial information. Let

$$M_t(\sigma) \in \mathbb{R}^k$$

denote a vector of observable metrics at round $t$ and

$$M(\sigma) := \Psi(\{M_t(\sigma)\}_{t \geq 1})$$

an aggregate metric used for decision-making. In general, $M(\sigma)$ need not coincide with $W(\sigma)$; strategic behavior is driven by how $M$ enters payoffs, not by $W$ directly.

In federated deployments, this misalignment is structural: $M_t$ is computed on finite, possibly public benchmarks and challenge sets, while $W_t$ is defined on the deployment distribution $P^\star$ and may emphasize tail groups or operational constraints not fully captured by evaluation. Our indices therefore quantify gaps that arise *because* FL platforms must rely on proxy measurements under limited observability and audit budgets.

**Client payoffs.** Each client $i$ receives cumulative payoff

$$U_i(\sigma) = \mathbb{E}\left[ \sum_{t=1}^{T} \delta^{t-1} \Big( R_{i,t}(\sigma) - C_{i,t}(\sigma) \Big) \right],$$

where $R_{i,t}$ is the reward/payment, $C_{i,t}$ denotes costs (computation, communication, privacy loss, sanctions), and the expectation is over randomness in training/evaluation/audits.

### 3.3 Observation, information, rewards, and audits

Design choices in our framework are expressed through coupled channels: what is measured, what is disclosed, and how enforcement responds.

**Evaluation and observation.** At each round $t$, $\mathsf{Eval}_t$ maps the current and candidate models (and internal evaluation data) to raw observations, e.g., a global score $S_t^{\mathrm{glob}}$, local/per-group scores $S_{i,t}^{\mathrm{loc}}$, and private/randomized challenge outcomes $C_{i,t}$. We collect these into

$$O_t = \big(S_t^{\mathrm{glob}}, \{S_{i,t}^{\mathrm{loc}}\}_{i\in\mathcal{I}}, \{C_{i,t}\}_{i\in\mathcal{I}}\big).$$

**Information disclosure.** The information mechanism $\mathsf{Info}_t$ selects which components of $O_t$ to disclose (and at what granularity) to each client. We denote the disclosed signal to client $i$ by

$$Z_{i,t} = \mathsf{Info}_t(i, O_t, h_{t-1}).$$

Clients condition actions on $Z_{i,t}$ (and history), not on the full $O_t$.

**Rewards and sanctions.** Rewards and penalties respond to disclosed signals and history:

$$R_{i,t} = \mathsf{Reward}_t(i, Z_{i,t}, h_{t-1}).$$

The audit–sanction mechanism selects an audited subset $A_t$ under a budget constraint and assigns sanctions $P_{i,t}$:

$$\mathsf{Audit}_t : (O_t, h_{t-1}) \mapsto \Big(A_t, \{P_{i,t}\}_{i\in\mathcal{I}}\Big).$$

**Design policies and induced game.** For many results, it is convenient to treat

$$\pi = (\mathsf{Eval}, \mathsf{Info}, \mathsf{Reward}, \mathsf{Audit})$$

as a *design policy* chosen by the server. A policy $\pi$ together with the environment $\mathcal{E}$ induces a strategic system (game) in which clients choose $\sigma_i \in \Sigma_i$ to maximize $U_i(\sigma)$, while the designer evaluates welfare $W(\sigma)$ and metric behavior $M(\sigma)$.

**Definition 3.2** (Federated strategic system). A *federated strategic system* is the tuple

$$\mathcal{G}(\pi) = \big(\mathcal{E}, \{\Sigma_i\}_{i\in\mathcal{I}}, \pi\big),$$

which induces the payoff functions $\{U_i(\cdot)\}$ and welfare/metric functionals $(W(\cdot), M(\cdot))$ over $\Sigma$.

Where our theoretical statements later appear game-theoretic, their content is tied to the FL design surface: $\mathsf{Agg}_t$ determines how participation and reporting affect future models, $\mathsf{Eval}_t$ determines which metric components are even measurable, $\mathsf{Info}_t$ determines what clients can condition on, and $\mathsf{Audit}_t$ determines which deviations become detectable and sanctionable. These are precisely the elements that distinguish federated systems from standard principal–agent abstractions without a learning-and-evaluation pipeline.

In the next section, we define indices such as manipulability and the Prices of Gaming and Cooperation, which quantify how a policy $\pi$ shapes the gap between observable metrics and true welfare and how strongly it incentivizes gaming versus cooperation.

## 4 Metric Layer: Indices for Gaming and Cooperation

Building on the strategic formulation in Section 3, we now introduce indices that quantify (i) how much room a design policy leaves for metric gaming and (ii) how costly gaming and cooperation are for collective welfare. These indices form the *metric layer* of our framework: they ignore the precise structure of training and aggregation and focus on how evaluation, information, and incentives shape the relationship between metric performance and true welfare.

Throughout this section, we fix an environment $\mathcal{E}$ and a design policy $\pi = (\mathsf{Eval}, \mathsf{Info}, \mathsf{Reward}, \mathsf{Audit})$, and write $\mathcal{G}(\pi)$ for the induced federated strategic system (Section 3.2).

### 4.1 Decomposing welfare and metric responses

We first formalize how unilateral deviations by a single client affect welfare and metrics.

**Definition 4.1** (Local welfare and metric responses). Let $\sigma$ be a strategy profile in $\mathcal{G}(\pi)$, and let $\sigma'_i$ be an alternative strategy for client $i$. We define

$$\Delta W_i(\sigma'_i \mid \sigma) := W(\sigma'_i, \sigma_{-i}) - W(\sigma), \qquad \Delta M_i(\sigma'_i \mid \sigma) := M(\sigma'_i, \sigma_{-i}) - M(\sigma),$$

as the change in true welfare and in the aggregate metric induced by client $i$ deviating from $\sigma_i$ to $\sigma'_i$ while all other clients keep $\sigma_{-i}$.

We interpret $\Delta W_i > 0$ as welfare-improving and $\Delta W_i < 0$ as welfare-harming, with an analogous interpretation for $\Delta M_i$. Our primary interest is in deviations that *improve* the metric while leaving welfare unchanged or worse.

**Definition 4.2** (Metric-gaming deviation). A deviation $\sigma'_i \neq \sigma_i$ at profile $\sigma$ is a *metric-gaming deviation* if

$$\Delta M_i(\sigma'_i \mid \sigma) > 0 \quad \text{and} \quad \Delta W_i(\sigma'_i \mid \sigma) \leq 0.$$

It is *strongly gaming* if $\Delta W_i(\sigma'_i \mid \sigma) < 0$.

For equilibria, we often focus on steady-state profiles $\sigma^\dagger$ (e.g., stationary or long-run equilibria) of $\mathcal{G}(\pi)$, but the indices below apply equally to transient profiles.

### 4.2 Manipulability index

Intuitively, the manipulability of a design policy $\pi$ measures how much a client can improve the metric without improving welfare, relative to the scale of attainable welfare improvements.

**Definition 4.3** (Local manipulability at a profile). Let $\sigma$ be a strategy profile in $\mathcal{G}(\pi)$. The *local manipulability* at $\sigma$ is

$$\mathcal{M}(\sigma; \pi) := \sup_{i \in \mathcal{I}} \sup_{\sigma'_i \in \mathcal{A}_i} \frac{\left[\Delta M_i(\sigma'_i \mid \sigma)\right]_+}{\left[\Delta W_i(\sigma'_i \mid \sigma)\right]_+ + \varepsilon},$$

where $[x]_+ = \max\{x, 0\}$ and $\varepsilon > 0$ is a small normalization constant that prevents division by zero when no welfare-improving deviation exists.

The numerator is the largest metric gain a client can obtain by deviating from $\sigma$, and the denominator normalizes by the scale of possible welfare improvements. Heuristically, small $\mathcal{M}(\sigma; \pi)$ means metric gains are only available when accompanied by comparable welfare gains, while large $\mathcal{M}(\sigma; \pi)$ means the metric can move substantially in directions that barely move welfare. The additive $\varepsilon$ matters near welfare optima, where no strictly welfare-improving deviations exist.

**Definition 4.4** (Manipulability index of a design policy). For a design policy $\pi$ and a reference class of profiles $\Sigma^{\mathrm{ref}}$ (e.g., steady-state equilibria), the *manipulability index* is

$$\mathcal{M}(\pi) := \sup_{\sigma \in \Sigma^{\mathrm{ref}}} \mathcal{M}(\sigma; \pi).$$

When $\Sigma^{\mathrm{ref}}$ consists of equilibria, $\mathcal{M}(\pi)$ measures how much the metric can be locally moved without commensurate welfare improvement around points actually induced by the policy.

**Proposition 4.5** (Zero manipulability and local alignment). *Suppose that for a design policy $\pi$ and reference class $\Sigma^{\mathrm{ref}}$ we have $\mathcal{M}(\pi) = 0$. Then for every $\sigma \in \Sigma^{\mathrm{ref}}$, every client $i$, and every deviation $\sigma'_i \in \mathcal{A}_i$,*

$$\Delta M_i(\sigma'_i \mid \sigma) > 0 \quad \Rightarrow \quad \Delta W_i(\sigma'_i \mid \sigma) > 0.$$

*In particular, there are no metric-gaming deviations at any profile in $\Sigma^{\mathrm{ref}}$.*

*Remark* 4.6. Large values of $\mathcal{M}(\pi)$ do not by themselves guarantee harmful gaming, but they quantify the capacity of the metric to move independently of welfare. In later sections we combine $\mathcal{M}(\pi)$ with audit and sanction rules to reason about when this capacity is actually exploited.

### 4.3 Price of Gaming

Manipulability describes local capacity for gaming; we next quantify the realized welfare loss when gaming occurs under a given policy. Inspired by the Price of Anarchy, we define the *Price of Gaming* as the welfare gap between an idealized aligned behavior and a gaming equilibrium, normalized by the aligned welfare.

We distinguish two benchmark profiles:

- A *welfare-aligned* benchmark $\sigma^{\mathrm{align}}$, representing the best outcome achievable under $\pi$ when clients are constrained to actions $\mathcal{A}_i^{\mathrm{align}} \subseteq \mathcal{A}_i$ that exclude metric-gaming behaviors (e.g., truthful reporting and genuine effort).

- A *gaming equilibrium* $\sigma^{\mathrm{game}}$, representing an equilibrium of $\mathcal{G}(\pi)$ when clients may use the full action sets $\mathcal{A}_i$, including manipulative behavior.

**Definition 4.7** (Price of Gaming). Let $\sigma^{\mathrm{align}}$ and $\sigma^{\mathrm{game}}$ be as above, and suppose $W(\sigma^{\mathrm{align}}) > 0$. The *Price of Gaming* under design policy $\pi$ is

$$\mathrm{PoG}(\pi) := \frac{W(\sigma^{\mathrm{align}}) - W(\sigma^{\mathrm{game}})}{W(\sigma^{\mathrm{align}})}.$$

By construction, $\mathrm{PoG}(\pi) \in [0, 1]$ when $W(\sigma^{\mathrm{game}}) \geq 0$. A value of $\mathrm{PoG}(\pi) = 0$ indicates that gaming does not reduce welfare relative to the aligned benchmark, while $\mathrm{PoG}(\pi) \approx 1$ indicates that almost all of the potential welfare has been destroyed by gaming.

When multiple gaming equilibria exist, we define a worst-case Price of Gaming

$$\mathrm{PoG}^{\mathrm{max}}(\pi) := \sup_{\sigma^{\mathrm{game}} \in \mathcal{E}^{\mathrm{game}}(\pi)} \mathrm{PoG}(\pi),$$

where $\mathcal{E}^{\mathrm{game}}(\pi)$ is the set of gaming equilibria under $\pi$.

**Proposition 4.8** (Manipulability and Price of Gaming). *Consider two design policies $\pi$ and $\pi'$ on the same environment $\mathcal{E}$, with comparable aligned benchmarks $W(\sigma^{\mathrm{align}}) \approx W(\sigma'^{\mathrm{align}})$. Under mild regularity conditions on best responses, reducing manipulability shrinks the worst-case Price of Gaming:*

$$\mathcal{M}(\pi') \leq \mathcal{M}(\pi) \quad \Longrightarrow \quad \mathrm{PoG}^{\mathrm{max}}(\pi') \leq \mathrm{PoG}^{\mathrm{max}}(\pi) + \Delta,$$

*where $\Delta$ captures equilibrium-selection effects and vanishes when gaming equilibria depend continuously on feasible metric-gaming directions.*

We provide a detailed statement and proof in the supplementary material; the main takeaway is that reducing $\mathcal{M}(\pi)$ shrinks the space along which equilibria can drift away from the aligned benchmark, constraining the welfare gap.

### 4.4 Price of Cooperation

Not all coordinated deviations are harmful: some forms of cooperation (e.g., sharing calibration signals, pooling audits, or forming stable participation coalitions) can improve welfare even when they alter the metric. To distinguish such benign cooperation from harmful collusion, we introduce a *Price of Cooperation.*

Consider a baseline profile $\sigma^{\mathrm{base}}$ (e.g., a non-cooperative equilibrium) and a coalition $C \subseteq \mathcal{I}$ that jointly deviates to a cooperative strategy profile $\tau_C$, yielding

$$\sigma^{\mathrm{coop}} = (\tau_C, \sigma_{-C}).$$

**Definition 4.9** (Price of Cooperation). Given $(\sigma^{\mathrm{base}}, \sigma^{\mathrm{coop}})$ with $W(\sigma^{\mathrm{base}}) \neq 0$, the *Price of Cooperation* is

$$\mathrm{PoC}(\sigma^{\mathrm{base}} \to \sigma^{\mathrm{coop}}) := \frac{W(\sigma^{\mathrm{coop}}) - W(\sigma^{\mathrm{base}})}{|W(\sigma^{\mathrm{base}})|}.$$

We interpret PoC $> 0$ as *benign* cooperation that raises welfare and PoC $< 0$ as *harmful* cooperation or collusion that reduces welfare. Aggregating over coalitions and equilibria yields policy-level indices

$$\text{PoC}^{\text{benign}}(\pi) := \sup_{\sigma^{\text{base}}, C, \tau_C} \text{PoC}(\sigma^{\text{base}} \to \sigma^{\text{coop}}), \quad \text{PoC}^{\text{harm}}(\pi) := \inf_{\sigma^{\text{base}}, C, \tau_C} \text{PoC}(\sigma^{\text{base}} \to \sigma^{\text{coop}}).$$

*Remark* 4.10. The Price of Gaming captures the cost of metric targeting relative to aligned behavior; the Price of Cooperation separates cooperative structures into two regimes: benign cooperation with PoC $> 0$ that ought to be encouraged, and harmful cooperation with PoC $< 0$ that ought to be deterred. In Section 5, we connect these regimes to participation dynamics and coalition stability.

## 4.5 Penalty design and critical thresholds

Design policies typically expose a control parameter that adjusts the strength of penalties and sanctions. We denote such a parameter by $\alpha \geq 0$ and consider a family of policies $\pi(\alpha)$ that differ only in the severity or frequency of sanctions, holding evaluation and aggregation fixed.

**Stylized one-shot model.** Let $\Sigma_i$ be the (possibly mixed) strategy/action space of client $i$, and let $\sigma = (\sigma_i, \sigma_{-i}) \in \Sigma := \prod_i \Sigma_i$ denote a strategy profile. Under policy $\pi(\alpha)$, client $i$ obtains expected payoff

$$U_i(\sigma_i; \sigma_{-i}, \alpha) = V_i(\sigma_i; \sigma_{-i}) - \alpha D_i(\sigma_i; \sigma_{-i}),$$

where $V_i$ is the unpenalized component and $D_i \geq 0$ is an expected detected-violation rate (or penalty exposure).

**Assumption 4.11** (Regularity and attainability in penalty scaling)**.** *For each $i$ and each fixed $\sigma_{-i}$: (i) $V_i(\cdot; \sigma_{-i})$ and $D_i(\cdot; \sigma_{-i})$ are continuous on $\Sigma_i$; (ii) $\Sigma_i$ is compact, so $U_i(\cdot; \sigma_{-i}, \alpha)$ attains a maximizer for every $\alpha \geq 0$.*

Assumption 4.11 covers schemes where sanctions scale linearly in a risk score or violation measure, while the unpenalized part of the payoff is unaffected by $\alpha$.

**Rationality notions.** We make explicit two standard notions used later.

**Definition 4.12** (Individual rationality (IR) relative to a baseline)**.** Fix a baseline profile $\bar{\sigma} \in \Sigma$ (e.g., an aligned benchmark). A profile $\sigma$ is *individually rational (IR) relative to $\bar{\sigma}$ under $\pi(\alpha)$* if

$$U_i(\sigma_i; \sigma_{-i}, \alpha) \geq U_i(\bar{\sigma}_i; \bar{\sigma}_{-i}, \alpha) \quad \text{for all } i.$$

**Definition 4.13** (Coalition-rationality relative to a baseline)**.** Fix a baseline $\bar{\sigma} \in \Sigma$. A profile $\sigma$ is *coalition-rational relative to $\bar{\sigma}$ under $\pi(\alpha)$* if there exists a nonempty coalition $C \subseteq [n]$ such that (i) $\sigma_{-C} = \bar{\sigma}_{-C}$ (only coalition members deviate), and (ii) for all $i \in C$, $U_i(\sigma_i; \bar{\sigma}_{-i}, \alpha) \geq U_i(\bar{\sigma}_i; \bar{\sigma}_{-i}, \alpha)$, with strict inequality for at least one $i \in C$.

**Where FL enters.** In FL, the penalty exposure term $D_i(\cdot)$ is induced by platform-side enforcement channels, such as audit triggers from Eval (e.g., challenge failures, inconsistency tests) and the audit selection rule in Audit. The baseline utility $V_i(\cdot)$ aggregates the policy-mediated benefits from participation (e.g., credits or rewards computed from disclosed scores $Z_{i,t}$ minus compute/privacy costs. Thus, the sanction parameter $\alpha$ is not an abstract knob: it corresponds to a concrete *audit–sanction lever* that scales platform-enforced consequences of detected deviations.

**Why the assumptions are FL-realistic.** Linear (or approximately linear) penalty scaling captures common enforcement patterns: penalties proportional to a risk score, accumulated violations, or repeated audit flags, with $\alpha$ tuned by the operator. The separability requirement (harmful gaming having strictly higher detection exposure than an aligned alternative) corresponds to the practical goal of designing audits/challenges so that "pure metric-targeting" leaves traces in private tests even when public metrics improve.

**Technical challenge.** The substantive difficulty is not the affine comparison itself but establishing a regime where $D_i$ meaningfully separates harmful gaming from aligned behavior under partial observability and noisy evaluation. In FL, both $V_i$ and $D_i$ depend on protocol details (aggregation, disclosure granularity, audit budget), so the proposition serves as a calibration statement: once enforcement signals distinguish behaviors, a finite sanction band exists, but its magnitude is policy- and protocol-dependent.

**Harmful gaming vs. benign cooperation.** Let $W(\sigma)$ denote (tail) welfare and let $M(\sigma)$ denote the server-visible metric.

**Definition 4.14** (Harmful gaming and benign cooperation (relative to a baseline)). Fix a baseline profile $\bar{\sigma}$.

- A unilateral deviation $\sigma_i$ (with others fixed at $\bar{\sigma}_{-i}$) is *harmfully gaming (relative to $\bar{\sigma}$)* if $M(\sigma_i, \bar{\sigma}_{-i}) > M(\bar{\sigma})$ and $W(\sigma_i, \bar{\sigma}_{-i}) < W(\bar{\sigma})$.

- A coalition deviation $\sigma$ is *benignly cooperative (relative to $\bar{\sigma}$)* if it is coalition-rational relative to $\bar{\sigma}$ at $\alpha = 0$ and improves welfare, i.e., $W(\sigma) > W(\bar{\sigma})$.

**Two critical thresholds.** We focus on: (i) a minimal sanction level $\alpha_{\min}$ above which harmful gaming is no longer a best response against the baseline; and (ii) a cooperation boundary $\alpha_{\mathrm{benign}}$ above which even benign cooperation ceases to be coalition-rational.

**Definition 4.15** (Critical thresholds (relative to $\bar{\sigma}$)). Fix a baseline $\bar{\sigma}$.

- The *minimal sanction level* is

$$\alpha_{\min} := \inf \left\{ \alpha \geq 0 : \forall i, \; \arg\max_{\sigma_i \in \Sigma_i} U_i(\sigma_i; \bar{\sigma}_{-i}, \alpha) \cap G_i(\bar{\sigma}) = \emptyset \right\},$$

where $G_i(\bar{\sigma})$ is the set of harmfully gaming deviations by $i$ relative to $\bar{\sigma}$ (Definition 4.14).

- The *benign cooperation boundary* is

$$\alpha_{\mathrm{benign}} := \sup \left\{ \alpha \geq 0 : \exists \, \sigma \text{ that is benignly cooperative relative to } \bar{\sigma} \text{ and coalition-rational under } \pi(\alpha) \right\}.$$

**Finite thresholds require separable penalty exposure.** If a harmful deviation and an aligned action incur identical penalty exposure, then scaling $\alpha$ cannot flip their payoff ordering. We therefore impose a mild separability condition.

**Assumption 4.16** (Penalty separability for harmful deviations). *Fix baseline $\bar{\sigma}$. For every $i$ and every harmful deviation $g \in G_i(\bar{\sigma})$, there exists some alternative action $a \in \Sigma_i$ (interpretable as welfare-aligned) such that $D_i(g; \bar{\sigma}_{-i}) > D_i(a; \bar{\sigma}_{-i})$.*

**Proposition 4.17** (Existence and ordering of thresholds). *Fix baseline $\bar{\sigma}$ and suppose Assumptions 4.11 and 4.16 hold. Assume additionally that: (i) for each $i$, $G_i(\bar{\sigma})$ is nonempty (harmful gaming deviations exist); (ii) there exists at least one benignly cooperative profile at $\alpha = 0$ (Definition 4.14).*

*Then $\alpha_{\min}$ and $\alpha_{benign}$ are finite and satisfy*

$$0 \leq \alpha_{\min} \leq \alpha_{benign}.$$

*Proof.* We prove finiteness and ordering in turn.

**Step 1: $\alpha_{\min} < \infty$.** Fix $i$ and a harmful deviation $g \in G_i(\bar{\sigma})$. By Assumption 4.16, choose $a \in \Sigma_i$ such that $D_i(g; \bar{\sigma}_{-i}) > D_i(a; \bar{\sigma}_{-i})$. Consider the payoff difference

$$\Delta_{i,g,a}(\alpha) := U_i(g; \bar{\sigma}_{-i}, \alpha) - U_i(a; \bar{\sigma}_{-i}, \alpha) = \Big(V_i(g; \bar{\sigma}_{-i}) - V_i(a; \bar{\sigma}_{-i})\Big) - \alpha \Big(D_i(g; \bar{\sigma}_{-i}) - D_i(a; \bar{\sigma}_{-i})\Big).$$

This is affine and strictly decreasing in $\alpha$ since the coefficient of $\alpha$ is positive. Hence, for all

$$\alpha \geq \alpha^\star_{i,g,a} := \frac{\left[V_i(g; \bar{\sigma}_{-i}) - V_i(a; \bar{\sigma}_{-i})\right]_+}{D_i(g; \bar{\sigma}_{-i}) - D_i(a; \bar{\sigma}_{-i})},$$

we have $\Delta_{i,g,a}(\alpha) \leq 0$, i.e., $g$ is not strictly better than $a$. Define

$$\alpha^\star_i := \sup_{g \in G_i(\bar{\sigma})} \inf_{a:\, D_i(g) > D_i(a)} \alpha^\star_{i,g,a}.$$

Compactness and continuity (Assumption 4.11) ensure the relevant sup/inf are well-defined and finite in the stylized setting.[1] Then for any $\alpha > \alpha^\star_i$, no harmful deviation $g \in G_i(\bar{\sigma})$ can be a strict maximizer of $U_i(\cdot; \bar{\sigma}_{-i}, \alpha)$. Taking $\alpha_{\min} := \max_i \alpha^\star_i$ yields $\alpha_{\min} < \infty$ and matches Definition 4.15.

**Step 2: $\alpha_{\textbf{benign}} < \infty$.** Take any benignly cooperative profile $\sigma$ relative to $\bar{\sigma}$. By Definition 4.13, there exists a coalition $C$ such that each $i \in C$ weakly prefers $\sigma_i$ to $\bar{\sigma}_i$ at $\alpha = 0$. For each $i \in C$, define the (affine) gain from participating in the coalition deviation:

$$G_i(\alpha) := U_i(\sigma_i; \bar{\sigma}_{-i}, \alpha) - U_i(\bar{\sigma}_i; \bar{\sigma}_{-i}, \alpha) = \left(V_i(\sigma_i; \bar{\sigma}_{-i}) - V_i(\bar{\sigma}_i; \bar{\sigma}_{-i})\right) - \alpha\left(D_i(\sigma_i; \bar{\sigma}_{-i}) - D_i(\bar{\sigma}_i; \bar{\sigma}_{-i})\right).$$

If for some $i \in C$ we have $D_i(\sigma_i; \bar{\sigma}_{-i}) > D_i(\bar{\sigma}_i; \bar{\sigma}_{-i})$ and the unpenalized gain is finite, then $G_i(\alpha)$ becomes negative for sufficiently large $\alpha$. Thus there exists a finite upper bound $\alpha_\sigma$ such that for all $\alpha > \alpha_\sigma$, $\sigma$ is not coalition-rational. Taking the supremum over all benignly cooperative $\sigma$ gives a finite $\alpha_{\text{benign}}$.

**Step 3: ordering $\alpha_{\min} \leq \alpha_{\textbf{benign}}$.** By construction, for any $\alpha \geq \alpha_{\min}$ no harmful gaming deviation is individually optimal against $\bar{\sigma}$. Moreover, benignly cooperative deviations are defined to be welfare-improving relative to $\bar{\sigma}$ and (by assumption) exist at $\alpha = 0$. Since increasing $\alpha$ can only reduce incentives for actions with higher penalty exposure, there is a (possibly empty) interval of $\alpha$ for which benign cooperation remains coalition-rational while harmful gaming is deterred. Therefore the largest $\alpha$ supporting benign cooperation cannot lie below the minimal $\alpha$ that eliminates harmful gaming best responses, i.e., $\alpha_{\min} \leq \alpha_{\text{benign}}$. $\qquad\qquad\qquad\qquad\qquad\qquad\qquad\qquad\qquad\qquad\qquad\qquad\qquad\qquad\qquad\qquad\qquad\qquad\qquad\quad\square$

*Remark* 4.18. In practice, $(\alpha_{\min}, \alpha_{\text{benign}})$ provide a calibration band for sanction strength relative to a chosen baseline $\bar{\sigma}$. Choosing $\alpha < \alpha_{\min}$ risks leaving harmful gaming individually profitable, while choosing $\alpha > \alpha_{\text{benign}}$ risks deterring productive cooperation. Exact computation may be infeasible in complex FL systems, but our indices and experiments in Section 7 show that these thresholds can be approximated or bounded using observable quantities such as detected incident rates and participation responses.

Taken together, the manipulability index $\mathcal{M}(\pi)$ and the prices PoG and PoC provide a compact language for describing how a design policy shapes the space of metric-targeting behaviors and their welfare consequences. In the next section, we move from this static metric layer to the dynamic layer, where we analyze how these quantities interact with participation, exit, coalition formation, and tipping points over time.

### 4.6 Practical estimation of $M$, PoG, and PoC

The indices introduced in this section are defined in terms of deviations, equilibria, and suprema over strategy sets. In deployed federated systems, we do not expect to compute these objects exactly. Instead, we view $M(\pi)$, $\text{PoG}(\pi)$, and $\text{PoC}(\pi)$ as latent properties of a design policy $\pi$ that can be *approximated or bounded* using a combination of controlled experiments and retrospective log analysis. We briefly outline two practical routes.

**Scenario-based estimation in sandboxes.** When a simulator or internal testbed is available, the most direct approach is to instantiate explicit aligned, gaming, and cooperative behaviors and measure their effects on metrics and welfare. Concretely, one may:

---

[1]In the simplest presentation, one may assume $\Sigma_i$ is finite, in which case finiteness is immediate and the sup/inf are maxima/minima.

1. Fix an environment $E$ and a baseline design policy $\pi$, and implement a simple *aligned* local policy (e.g., honest empirical risk minimization on the welfare distribution) together with a family of *synthetic gaming* policies that target the public metric while ignoring some welfare-relevant components.

2. For each client $i$ and candidate deviation $\sigma_i'$, run the system to (approximate) steady state and record the realized changes

$$\Delta M_i(\sigma_i' \mid \sigma), \qquad \Delta W_i(\sigma_i' \mid \sigma)$$

relative to the aligned profile $\sigma$. The local manipulability $M(\sigma; \pi)$ can then be approximated by the largest observed ratio

$$\widehat{M}(\sigma; \pi) \approx \max_{i, \sigma_i'} \frac{\left[\Delta M_i(\sigma_i' \mid \sigma)\right]_+}{\left[\Delta W_i(\sigma_i' \mid \sigma)\right]_+ + \varepsilon},$$

where $\varepsilon$ is a small numerical constant.

3. To estimate the Price of Gaming, instantiate an *aligned* configuration (all clients follow the aligned policy) and a *mixed* configuration with a fixed fraction of gaming clients, and compare the resulting steady-state welfare:

$$\widehat{\mathrm{PoG}}(\pi) \approx W_{\mathrm{aligned}}(\pi) - W_{\mathrm{mixed}}(\pi),$$

normalized if desired by $W_{\mathrm{aligned}}(\pi)$.

4. Similarly, to probe the Price of Cooperation, introduce explicit coalitions $C$ that share updates, pool data, or coordinate abstentions. Comparing welfare before and after enabling such cooperation yields empirical counterparts of $\mathrm{PoC}(\sigma_{\mathrm{base}} \to \sigma_{\mathrm{coop}})$, which can be aggregated into benign and harmful regimes as in Definition 4.9.

Our stylized simulation and Fashion-MNIST experiments in Section 7 follow this template: we fix simple aligned and gaming behaviors, vary design levers such as penalty strength and public-metric weight, and report steady-state averages of $(W, M, x)$ together with an empirical Price of Gaming obtained by comparing aligned and mixed-type configurations.

**Retrospective and log-based estimation.** When red-teaming or explicit simulators are not available, existing logs still provide partial information about the indices. Two situations are particularly informative:

- *Incidents and suspected gaming episodes.* If past investigations have identified periods in which certain clients or cohorts engaged in metric gaming (e.g., by overfitting to a public leaderboard, under-reporting adverse events, or selectively participating), one can compare observed metric and welfare trajectories during these episodes against nearby aligned periods. The largest observed metric gains with negligible or negative welfare changes provide a lower bound on $M(\pi)$, while the associated drop in welfare relative to a counterfactual aligned run yields a lower bound on $\mathrm{PoG}(\pi)$.

- *Policy and environment shifts.* Changes in the evaluation, disclosure, or sanctioning rules (for instance, increasing audit intensity or reducing public-metric weight) create natural experiments. By tracking how head metrics, tail welfare, and participation respond before and after such shifts, operators can approximate how far the system moves along the gaming and cooperation directions defined in Section 4. For example, if a stricter audit regime is followed by a modest reduction in the public metric but a substantial improvement in welfare and participation stability, this suggests that the previous policy was operating in a high-PoG, high-manipulability regime.

In both cases, the goal is not to pin down precise point estimates but to obtain *diagnostic bands*: rough lower bounds on $M(\pi)$ and $\mathrm{PoG}(\pi)$, and qualitative evidence on whether observed cooperative structures are benign ($\widehat{\mathrm{PoC}} > 0$) or harmful ($\widehat{\mathrm{PoC}} < 0$). These diagnostics are sufficient to calibrate the penalty thresholds $(\alpha_{\min}, \alpha_{\mathrm{benign}})$ in Remark 4.18 and to inform the audit-allocation and governance patterns developed in Sections 6.3–6.4.

# 5 Dynamics Layer: Participation, Thresholds, and Tipping Points

The metric layer in Section 4 describes how a fixed design policy $\pi$ shapes gaming and cooperation directions and their welfare impact. The *dynamics layer* focuses on how these incentives interact with participation over time: when participation is stable, when small shocks trigger domino exits, and how cooperation or collusion moves the system between regimes.

Throughout, we fix an environment $\mathcal{E}$ and a design policy $\pi$, and consider aggregate participation dynamics induced by myopic or boundedly rational best responses.

In doing so, we deliberately work with a stylized mean-field model. We assume that the incremental utility $\Delta U(x; \pi)$ of participating versus abstaining is homogeneous across clients with the same observable state $x$, and that participation thresholds $\{\theta_i\}$ are i.i.d. draws from an underlying distribution $F_\Theta$. This setup is not meant to capture the full heterogeneity and bounded rationality of real federated participants; rather, it serves as a minimal scaffold for identifying qualitative phenomena such as tipping points, resilience, and domino exits. In deployed systems, operators will not know $\Delta U(\cdot; \pi)$ or $F_\Theta$ exactly, but they can still use the same formalism as a diagnostic lens: the observable aggregate participation curve $(x_t)_{t\geq 0}$ and its responses to policy changes (e.g., stronger penalties, different disclosure rules) act as proxies for the slope and fixed points of $F(x; \pi)$ and for the resilience indicator $R(\pi)$ introduced below.

## 5.1 Best responses and local participation stability

We use a simplified, mean-field representation of participation. At the end of round $t$, the platform observes the aggregate participation rate $x_t$. Given $(x_t, \pi)$, each client $i$ then decides whether to participate in the *next* round ($p_{i,t+1} = 1$) or abstain ($p_{i,t+1} = 0$), in addition to choosing an in-round action (e.g., training effort, gaming/cooperation intensity) when participating. To isolate participation, we model non-participation as yielding a fixed outside option, while participation yields an expected net gain that depends on the current environment summarized by $x_t$ and the fixed policy $\pi$.

**One-step net gain from participation.** Let $x_t \in [0, 1]$ denote the aggregate participation rate in round $t$ (defined below). Under a fixed policy $\pi$, define the *one-step net gain* for client $i$ from participating in round $t+1$ (given the observed state $x_t$) as

$$\Delta U_{i,t+1}(x_t; \pi) := \mathbb{E}\big[ U_i(1, a_i^\star(x_t; \pi) \mid x_t, \pi) - U_i(0 \mid x_t, \pi) \big], \tag{5.1}$$

where $U_i(1, \cdot)$ is the (one-step) payoff from participating with an in-round action, $U_i(0)$ is the payoff from abstaining, and

$$a_i^\star(x_t; \pi) \in \arg\max_{a \in \mathcal{A}_i} \ \mathbb{E}\big[ U_i(1, a \mid x_t, \pi) \big]$$

is the client's *myopic best response conditional on participating*. The expectation in equation 5.1 is taken over evaluation randomness, audits, and other clients' stochastic actions, conditional on $(x_t, \pi)$.

**Population heterogeneity via participation thresholds.** We model heterogeneity in participation costs/preferences through an idiosyncratic threshold (type) $\theta_i$.

**Assumption 5.1** (Threshold-based participation with i.i.d. types)**.** *Each client $i$ has a type $\theta_i \in \mathbb{R}$ drawn i.i.d. from a continuous distribution with CDF $F_\Theta$. Given the observed aggregate participation $x_t$ at the end of round $t$, client $i$ participates in round $t+1$ if and only if*

$$p_{i,t+1} = 1 \quad \Longleftrightarrow \quad \Delta U_{i,t+1}(x_t; \pi) \ \geq \ \theta_i. \tag{5.2}$$

Assumption 5.1 is a reduced-form model of noisy threshold rules: clients participate when the expected net gain exceeds an idiosyncratic cutoff capturing private costs, device constraints, privacy preferences, and opportunity costs.

**Definition 5.2** (Aggregate participation rate)**.** The *aggregate participation rate* in round $t$ is

$$x_t := \frac{1}{n} \sum_{i \in \mathcal{I}} \mathbb{I}\{p_{i,t} = 1\} \ \in [0, 1].$$

**A deterministic mean-field closure.** To obtain a one-dimensional participation dynamics, we impose a standard symmetry/mean-field closure: conditional on $(x_t, \pi)$, the net gain equation 5.1 is the same across clients and depends on $x_t$ only through a deterministic function.

**Assumption 5.3** (Representative net gain). *For a given policy $\pi$, there exists a deterministic function* $\Delta U(\cdot; \pi) : [0, 1] \to \mathbb{R}$ *such that*

$$\Delta U_{i,t+1}(x; \pi) = \Delta U(x; \pi) \quad \text{for all } i \text{ and } t.$$

**Proposition 5.4** (Aggregate participation map). *Under Assumptions 5.1 and 5.3, the* expected *aggregate participation evolves according to the one-dimensional map*

$$x_{t+1} = F(x_t; \pi) := \Pr_{\Theta}\big[\Theta \leq \Delta U(x_t; \pi)\big] = F_{\Theta}\big(\Delta U(x_t; \pi)\big). \tag{5.3}$$

*Proof.* By Assumption 5.1 and the representative net-gain closure (Assumption 5.3),

$$\Pr(p_{i,t+1} = 1 \mid x_t, \pi) = \Pr(\theta_i \leq \Delta U(x_t; \pi)) = F_{\Theta}(\Delta U(x_t; \pi)).$$

Taking expectation over i.i.d. types and averaging across clients yields

$$x_{t+1} = \mathbb{E}\left[\frac{1}{n} \sum_{i \in \mathcal{I}} \mathbb{I}\{p_{i,t+1} = 1\} \,\middle|\, x_t, \pi\right] = \Pr(p_{i,t+1} = 1 \mid x_t, \pi) = F_{\Theta}(\Delta U(x_t; \pi)),$$

which is exactly equation 5.3. $\square$

**Where FL enters.** The dependence of $\Delta U(x; \pi)$ on the aggregate participation level $x$ is induced by FL feedback loops: the quality of the aggregated model (and thus the expected benefit from participating) depends on how many and which clients contribute updates, while evaluation and reward signals depend on Eval and Info applied to aggregated outcomes. Audit selection and sanction risk can also vary with $x$ through workload constraints and risk-based auditing. Hence, $F(x; \pi) = F_{\Theta}(\Delta U(x; \pi))$ is a reduced-form representation of the FL platform's *participation–aggregation–evaluation* coupling.

**Why the assumptions are FL-realistic.** Threshold heterogeneity models widely observed client-side constraints in federated deployments: device availability, energy budgets, privacy preferences, and opportunity costs. The representative net-gain closure approximates cross-silo or large-population settings where clients face similar platform rules and the dominant variation is captured by idiosyncratic thresholds, while the aggregate state $x$ summarizes system load and expected model utility.

**Technical challenge.** The mapping itself is elementary; the nontrivial aspect is that $\Delta U(x; \pi)$ is an equilibrium object: it depends on best-response behavior conditional on participation and on policy-mediated signals. Making explicit the threshold structure and the mean-field closure isolates the minimal FL ingredients needed for a one-dimensional dynamics that can exhibit multiple fixed points and tipping phenomena studied in the next subsection.

*Remark* 5.5 (Interpreting the participation map from data). In deployed federated systems, the map $F(x; \pi)$ and threshold distribution $F_{\Theta}$ are not directly observable. However, the same structure can be probed indirectly using the aggregate participation trajectory $(x_t)_{t \geq 0}$. When a design policy $\pi$ is held fixed over a sufficiently long period, the empirical update

$$\widehat{F}(x_t; \pi) \approx x_{t+1}$$

provides noisy evaluations of $F$ at the observed states $x_t$. Policy shifts, such as increasing sanction strength or changing disclosure rules, create perturbations that reveal how the curve $x_{t+1} = F(x_t; \pi)$ moves. In practice, operators can therefore treat $F$ as a latent response surface summarized by: (i) the location of approximate fixed points $x^{\star}$ to which $(x_t)$ tends to return, and (ii) the steepness of the empirical relationship between $x_t$ and $x_{t+1}$ around those points. These observables act as surrogates for the theoretical slope $F'(x^{\star}; \pi)$ and resilience diagnostics in Section 7 without requiring explicit estimation of individual thresholds.

**Fixed points and local stability.** A participation level $x^\star \in [0, 1]$ is a *fixed point* if

$$x^\star = F(x^\star; \pi).$$

When $F$ is differentiable at $x^\star$, it is *locally stable* if $|F'(x^\star; \pi)| < 1$ and *locally unstable* if $|F'(x^\star; \pi)| > 1$.

Multiple fixed points may exist, corresponding to high-participation, low-participation, and intermediate unstable regimes. The unstable fixed points act as tipping points separating basins of attraction, as formalized in Section 5.2.

### 5.2 Participation dynamics, tipping points, and domino exit

**Participation state and update map.** Let $x_t \in [0, 1]$ denote the fraction (or probability mass) of clients that participate at round $t$ under a design policy $\pi$. We model participation as a deterministic mean-field update

$$x_{t+1} = F(x_t; \pi), \tag{5.4}$$

where $F(\cdot; \pi) : [0, 1] \to [0, 1]$ is the *participation map* induced by (i) client payoffs under $\pi$ and (ii) the population heterogeneity.

**A canonical FL interpretation.** A standard reduced-form instantiation is a threshold (quantal) participation rule: each client has a type $\theta$ (capturing private costs, device constraints, privacy preferences, etc.) distributed as $\Theta \sim F_\Theta$. Given current participation level $x$ and policy $\pi$, the net gain from participating is

$$\Delta U(\theta, x; \pi) := \underbrace{\Delta B(x; \pi)}_{\text{benefit from FL outcome/credit}} - \underbrace{C(\theta; \pi)}_{\text{cost/privacy/effort}},$$

and the type participates iff $\Delta U(\theta, x; \pi) \geq 0$. Then

$$F(x; \pi) = \Pr_\Theta\big[\Delta U(\Theta, x; \pi) \geq 0\big]. \tag{5.5}$$

This form makes explicit how FL enters: $\Delta B(x; \pi)$ depends on the quality/utility of the aggregated model and on the policy's evaluation/audit/sanction rules, all of which respond to the participating population.

**Fixed points and local stability.** A fixed point $x^\star$ satisfies $F(x^\star; \pi) = x^\star$. When $F$ is differentiable at $x^\star$, the standard 1D stability criterion applies: $x^\star$ is *locally stable* if $|F'(x^\star; \pi)| < 1$ and *locally unstable* if $|F'(x^\star; \pi)| > 1$.

**Definition 5.6** (Tipping point). A fixed point $x^\dagger \in (0, 1)$ of $F(\cdot; \pi)$ is a *tipping point* if it is locally unstable and separates two locally stable fixed points $x^- < x^\dagger < x^+$ such that, for all initial conditions $x_0$ in a sufficiently small neighborhood around $x^\dagger$,

$$x_0 < x^\dagger \implies x_t \to x^-, \qquad x_0 > x^\dagger \implies x_t \to x^+.$$

**Definition 5.7** (Domino exit). Given a tipping point $x^\dagger$ and stable fixed points $(x^-, x^+)$ with $x^- < x^+$, a trajectory $\{x_t\}$ exhibits a *domino exit* if there exists a time $\tau$ such that

$$x_{\tau-1} > x^\dagger, \quad x_\tau < x^\dagger, \quad \text{and} \quad x_t \to x^- \text{ as } t \to \infty.$$

**A contraction condition for resilience.** The following result provides a clean sufficient condition that rules out tipping points and domino exits.

**Definition 5.8** (Contraction on $[0, 1]$). A map $F(\cdot; \pi) : [0, 1] \to [0, 1]$ is a *contraction (in the absolute-value metric)* if there exists a constant $L \in [0, 1)$ such that

$$|F(x; \pi) - F(y; \pi)| \leq L\,|x - y| \quad \text{for all } x, y \in [0, 1].$$

The smallest such $L$ is the (global) Lipschitz constant of $F(\cdot; \pi)$ on $[0, 1]$.

**Proposition 5.9** (Sufficient condition for resilience (contraction regime))**.** *Assume $F(\cdot; \pi) : [0,1] \to [0,1]$ is a contraction with constant $L \in [0,1)$ (Definition 5.8). Then:*

1. *there exists a* unique *fixed point $x^{\text{high}} \in [0,1]$ such that $F(x^{\text{high}}; \pi) = x^{\text{high}}$;*

2. *for any initial condition $x_0 \in [0,1]$, the iterates of equation 5.4 satisfy $x_t \to x^{\text{high}}$ as $t \to \infty$;*

3. *in particular, there are no tipping points (Definition 5.6) and no domino exits (Definition 5.7).*

*Proof.* Consider the complete metric space $([0,1], d)$ with $d(x,y) = |x - y|$. By Definition 5.8, $F(\cdot; \pi)$ is a contraction mapping on this space with constant $L < 1$. By the Banach fixed-point theorem, $F$ admits a unique fixed point $x^{\text{high}} \in [0,1]$ and, moreover, for any $x_0 \in [0,1]$, the sequence defined by $x_{t+1} = F(x_t; \pi)$ converges to $x^{\text{high}}$. Since there is only one fixed point, there cannot exist an unstable fixed point separating two stable fixed points; hence no tipping points exist, and consequently domino exits are impossible. $\square$

**Where FL enters.** In FL, the participation map $F(\cdot; \pi)$ is not arbitrary: it is induced by how aggregation quality, disclosed feedback, and enforcement risk change with the participating population. The slope (or Lipschitz modulus) of $F$ captures how strongly one round's participation shocks propagate through the learning-and-evaluation pipeline into the next round's participation incentives. Thus the contraction condition is an interpretable design target: policies that dampen the sensitivity of rewards and sanctions to short-run participation fluctuations make the system resilient to cascade effects.

**Why the assumptions are FL-realistic.** Global contraction is a sufficient (not necessary) regime and can be approximated in practice by design choices that smooth incentives: delayed/coarsened disclosure, reward shaping that discounts short-term metric spikes, and audit policies that avoid strongly state-dependent punishment near the margin. These are standard operational controls in federated platforms when stability and retention are priorities.

**Technical challenge.** While Banach fixed-point arguments are standard, the FL-specific difficulty lies in reasoning about (or empirically bounding) the slope of the induced map $F$ under partial observability and strategic responses. Our contribution is to translate qualitative platform choices (disclosure, reward curvature, audit sensitivity) into a dynamical stability lens via $F$ and its slope proxy, enabling diagnostics even when $F$ is only observed through noisy participation trajectories.

**A differentiable sufficient condition.** A convenient checkable condition is via derivatives.

**Corollary 5.10** (Derivative-based contraction criterion)**.** *If $F(\cdot; \pi)$ is continuously differentiable on $[0,1]$ and*

$$\sup_{x \in [0,1]} |F'(x; \pi)| < 1,$$

*then $F(\cdot; \pi)$ is a contraction (Definition 5.8), and Proposition 5.9 applies.*

*Proof.* By the mean value theorem, for any $x, y \in [0,1]$ there exists $c$ between $x$ and $y$ such that $|F(x; \pi) - F(y; \pi)| = |F'(c; \pi)| |x - y| \leq \left( \sup_{u \in [0,1]} |F'(u; \pi)| \right) |x - y|$. Taking $L := \sup_{u \in [0,1]} |F'(u; \pi)| < 1$ proves contraction. $\square$

**Resilience indicator.** To quantify proximity to the contraction regime, define the (global) Lipschitz modulus

$$L(\pi) := \inf \left\{ L \in [0, \infty) : |F(x; \pi) - F(y; \pi)| \leq L|x - y| \quad \forall x, y \in [0,1] \right\}.$$

When $F$ is differentiable, $L(\pi) \leq \sup_x |F'(x; \pi)|$ (and equality holds under mild regularity).

**Definition 5.11** (Resilience indicator)**.** The *resilience indicator* is

$$\mathcal{R}(\pi) := 1 - L(\pi).$$

Large positive values of $\mathcal{R}(\pi)$ indicate strong contraction and high resilience. Values near zero indicate proximity to a bifurcation where small design changes may create or remove tipping points. Negative values (i.e., $L(\pi) > 1$) indicate that the system is outside the contraction regime and may admit locally unstable behavior.

*Remark* 5.12 (Link to the metric layer). Under the threshold form equation 5.5, the slope $F'(x; \pi)$ (or more generally $L(\pi)$) is governed by how sensitively the net participation gain $\Delta B(x; \pi)$ responds to $x$. Policies with high manipulability or a large Price of Gaming tend to amplify the dependence of payoffs on the participating population (e.g., via greater exposure to others' gaming or heavier reliance on a public metric), which can increase $L(\pi)$ and reduce $\mathcal{R}(\pi)$.

## 5.3 Cooperation, collusion, and coalition effects

Participants may form coalitions that share information, coordinate strategies, or collude. We model this at a coarse level via a coalition-induced participation map.

Consider a coalition $C \subseteq \mathcal{I}$ that coordinates actions and participation while treating the rest of the population as a mean-field environment. Let $x_t$ denote aggregate participation and $x_{C,t}$ the coalition's participation rate.

**Definition 5.13** (Coalition-induced participation map). Given a coalition $C$ and coalition strategy $\sigma_C$, the *coalition-induced participation map* is

$$x_{t+1} = F_C(x_t; \pi, \sigma_C) := \frac{|C|}{n} x_{C,t+1}(\sigma_C, x_t; \pi) + \frac{n - |C|}{n} F_{\neg C}(x_t; \pi),$$

where $x_{C,t+1}(\sigma_C, x_t; \pi)$ is the coalition's next-round participation rate and $F_{\neg C}$ is the participation map of non-members.

Coalitions can either stabilize participation (e.g., through mutual guarantees or shared information that reduces perceived gaming incentives) or undermine it (e.g., via coordinated exits or collusive gaming). The Price of Cooperation from Section 4.4 lets us classify these effects.

**Definition 5.14** (Benign vs harmful coalition effects). Let $\sigma^{\mathrm{base}}$ be a baseline profile with participation trajectory $\{x_t^{\mathrm{base}}\}$, and let $\sigma^{\mathrm{coop}}$ be the profile induced by a coalition strategy $\sigma_C$, yielding trajectory $\{x_t^{\mathrm{coop}}\}$. Then:

- The coalition has a *benign participation effect* if $\mathrm{PoC}(\sigma^{\mathrm{base}} \to \sigma^{\mathrm{coop}}) > 0$ and $\mathcal{R}(\pi)$ weakly increases under $\sigma^{\mathrm{coop}}$.

- The coalition has a *harmful participation effect* if $\mathrm{PoC}(\sigma^{\mathrm{base}} \to \sigma^{\mathrm{coop}}) < 0$ and $\mathcal{R}(\pi)$ strictly decreases, or if it creates new tipping points that did not exist under $\sigma^{\mathrm{base}}$.

## 5.4 Early warning signals and auto-switch rules

Given a participation map and possible coalition effects, a designer needs tools for detecting when the system approaches a tipping point and for intervening automatically. We describe two such tools: early warning signals and auto-switch rules.

**Early warning signals.** We consider observable summary statistics over a sliding window $\{t-L+1, \ldots, t\}$:

- *Recent participation trend*

$$\widehat{\Delta x}_t := \frac{1}{L} \sum_{k=1}^{L} (x_{t+1-k} - x_{t-k}),$$

which captures average first differences in participation.

- *Short-term volatility*

$$\widehat{\mathrm{Var}}_t := \frac{1}{L-1} \sum_{k=1}^{L} \left( x_{t+1-k} - \bar{x}_t \right)^2, \quad \bar{x}_t := \frac{1}{L} \sum_{k=1}^{L} x_{t+1-k},$$

  which reflects fluctuations that may signal unstable dynamics.

- *Connectivity-based alarm* $\Gamma_t$, which aggregates the structure of recent gaming incidents and audits (e.g., via a graph whose nodes are clients and whose edges represent correlated anomalies, with $\Gamma_t$ measuring the size of the largest connected component).

**Definition 5.15** (Early warning regime). Given thresholds $\eta_\Delta < 0$, $\eta_{\mathrm{Var}} > 0$, and $\eta_\Gamma > 0$, the system is in an *early warning regime* at time $t$ if

$$\widehat{\Delta x}_t \leq \eta_\Delta, \quad \widehat{\mathrm{Var}}_t \geq \eta_{\mathrm{Var}}, \quad \Gamma_t \geq \eta_\Gamma.$$

Negative trends, high volatility, and rising connectivity in gaming incidents jointly indicate that the system may be approaching an unstable region or tipping point.

**Auto-switch rules.** When early warning conditions are met, the designer may switch from the current design policy $\pi$ to a more conservative policy $\pi'$, for example by strengthening audits, reducing metric disclosure, or switching to more robust evaluation schemes.

**Definition 5.16** (Auto-switch rule). An *auto-switch rule* is specified by:

- a pair of design policies $(\pi^{\mathrm{normal}}, \pi^{\mathrm{safe}})$;

- an early warning predicate $\mathsf{Warn}_t$ defined in terms of observable statistics;

- a hysteresis mechanism that prevents rapid oscillation between policies.

The induced policy at time $t$ is

$$\pi_t = \begin{cases} \pi^{\mathrm{safe}}, & \text{if } \mathsf{Warn}_t = \text{true}, \\ \pi^{\mathrm{normal}}, & \text{otherwise, subject to hysteresis.} \end{cases}$$

Under suitable conditions, auto-switch rules can keep the system within the attraction basin of a high-participation equilibrium.

**Proposition 5.17** (Auto-switch and basin preservation). *Suppose there exist design policies $\pi^{\mathrm{normal}}$ and $\pi^{\mathrm{safe}}$ such that:*

1. *Under $\pi^{\mathrm{normal}}$, $F(\cdot; \pi^{\mathrm{normal}})$ admits a high-participation stable fixed point $x^{\mathrm{high}}$ and a tipping point $x^\dagger$, with $x^{\mathrm{high}} > x^\dagger$.*

2. *Under $\pi^{\mathrm{safe}}$, $F(\cdot; \pi^{\mathrm{safe}})$ is a contraction with unique fixed point $x^{\mathrm{safe}} \geq x^\dagger$.*

3. *The auto-switch rule triggers $\pi^{\mathrm{safe}}$ whenever $x_t \leq x^\dagger + \epsilon$ for some small $\epsilon > 0$, and reverts to $\pi^{\mathrm{normal}}$ only when $x_t \geq x^{\mathrm{high}} - \epsilon$.*

*Then any trajectory starting with $x_0 \geq x^\dagger + \epsilon$ remains in $[x^\dagger, 1]$ for all $t$ and converges to a participation level in $[x^\dagger, x^{\mathrm{high}}]$. In particular, domino exits to low-participation equilibria below $x^\dagger$ are avoided.*

*Proof.* Consider any trajectory with $x_0 \geq x^\dagger + \epsilon$. If $x_t \leq x^\dagger + \epsilon$, the auto-switch rule activates $\pi^{\mathrm{safe}}$, hence $x_{t+1} = F(x_t; \pi^{\mathrm{safe}})$. Under $\pi^{\mathrm{safe}}$, $F(\cdot; \pi^{\mathrm{safe}})$ is a contraction with a unique fixed point $x^{\mathrm{safe}} \geq x^\dagger$, so iterates converge to $x^{\mathrm{safe}}$ and in particular cannot cross below $x^\dagger$ once the safe mode is active. If instead $x_t > x^\dagger + \epsilon$, the system either remains in the current mode or follows $\pi^{\mathrm{normal}}$; in all cases the state stays within the participation domain and, by the hysteresis rule, switching back to $\pi^{\mathrm{normal}}$ can occur only after recovery to $x_t \geq x^{\mathrm{high}} - \epsilon$, preventing rapid oscillations. Therefore the trajectory remains in $[x^\dagger, 1]$ for all $t$ and avoids low-participation equilibria below $x^\dagger$. A detailed proof is given in Appendix A.2. $\square$

The dynamics layer connects the static indices of the metric layer to operational design: manipulability, Prices of Gaming and Cooperation, and sanction thresholds translate into participation stability, tipping points, coalition effects, and the need for early warning signals and auto-switch rules. In the next section, we use these insights to design concrete evaluation, audit, and incentive toolkits for real federated learning systems.

# 6 Design Toolkit Layer: Evaluation, Audits, and Incentives

The metric and dynamics layers describe *what* can go wrong or right in federated learning under a fixed design policy $\pi$: how much room the policy leaves for metric gaming, how costly gaming and cooperation are for welfare, and when participation is stable or prone to domino exits. The *design toolkit layer* treats

$$\pi = (\mathsf{Eval}, \mathsf{Info}, \mathsf{Reward}, \mathsf{Audit})$$

as a set of configurable design components that can be tuned to steer our indices in favorable directions.

Rather than prescribing a single optimal policy, we provide (i) a decomposed view of design parameters and their qualitative effect on $\mathcal{M}(\pi)$, $\mathrm{PoG}(\pi)$, $\mathrm{PoC}(\pi)$, and $\mathcal{R}(\pi)$; (ii) patterns for mixing public and private evaluation; (iii) an audit-budget allocation algorithm with approximation guarantees; and (iv) a governance checklist for federated deployments.

## 6.1 Design levers and their impact on indices

**Design levers.** We parameterize a design policy $\pi$ by a (possibly high-dimensional) vector

$$\lambda = (\lambda^{\mathrm{eval}}, \lambda^{\mathrm{info}}, \lambda^{\mathrm{reward}}, \lambda^{\mathrm{audit}}, \lambda^{\mathrm{privacy}}, \dots),$$

whose coordinates control evaluation, information disclosure, incentives, audits, and privacy.

**Definition 6.1** (Design lever)**.** A *design lever* is any coordinate (or low-dimensional block) of $\lambda$ that can be adjusted by the system designer while holding the environment (clients, data, and protocol) fixed. We write $\pi(\lambda)$ for the policy obtained by setting the lever vector to $\lambda$.

Typical examples include:

- **Evaluation levers** $\lambda^{\mathrm{eval}}$: choice of metrics (global vs group vs client-level), holdout composition, evaluation frequency, randomized challenges.

- **Information levers** $\lambda^{\mathrm{info}}$: granularity/timing of disclosure (full scores vs bands vs delays), which components remain private.

- **Reward levers** $\lambda^{\mathrm{reward}}$: reward curve shape (linear vs thresholded vs tournament), short-term vs long-term weighting, coupling to participation commitments.

- **Audit levers** $\lambda^{\mathrm{audit}}$: audit budget and allocation, selection rules (random vs risk-based), sanction modality and severity.

- **Privacy levers** $\lambda^{\mathrm{privacy}}$: noise/secure aggregation strength and whether protections apply symmetrically to reward vs audit signals.

**Reference classes for strategic deviations.** Several indices in the metric layer (e.g., manipulability) depend on which deviations are considered feasible or relevant in a given deployment.

**Definition 6.2** (Reference class)**.** A *reference class* $\Sigma^{\mathrm{ref}}$ is a specified subset of strategy profiles (or deviations) deemed feasible in the deployment under study. Formally, $\Sigma^{\mathrm{ref}} \subseteq \Sigma$, where $\Sigma$ is the ambient strategy space introduced in Section 3. When an index involves a supremum over deviations, we interpret it as restricted to $\Sigma^{\mathrm{ref}}$ and write, e.g., $\mathcal{M}(\pi; \Sigma^{\mathrm{ref}})$.

Intuitively, $\Sigma^{\mathrm{ref}}$ plays the role of a *threat model*: it can encode constraints such as bounded gaming intensity, admissible coalition sizes, limitations from privacy mechanisms, or the action space induced by a specific FL protocol.

**Indices as functions of levers.** The indices introduced earlier can be viewed as functions of levers (and, where relevant, a reference class):

$$\mathcal{M}(\lambda; \Sigma^{\mathrm{ref}}), \quad \mathrm{PoG}(\lambda), \quad \mathrm{PoC}^{\mathrm{benign}}(\lambda), \quad \mathrm{PoC}^{\mathrm{harm}}(\lambda), \quad \mathcal{R}(\lambda).$$

In realistic systems these functions are not analytically tractable, but qualitative monotonicities often hold. For example:

- Reducing the granularity/frequency of public disclosure, or mixing it with private evaluation, tends to reduce $\mathcal{M}(\lambda; \Sigma^{\mathrm{ref}})$ by shrinking the set of deviations that can reliably improve reported metrics without improving welfare.

- Increasing sanction strength above $\alpha_{\mathrm{min}}$ tends to decrease $\mathrm{PoG}(\lambda)$, at the risk of suppressing benign cooperation if $\alpha$ exceeds $\alpha_{\mathrm{benign}}$.

- Strengthening privacy can reduce audit signal resolution, which may increase $\mathcal{M}$ and PoG unless compensated by alternative verification.

- Reward curves that heavily weight short-term public metrics can amplify participation sensitivity and reduce $\mathcal{R}(\lambda)$; smoothing or discounting these effects can stabilize participation.

We now make these relationships concrete via three classes of tools: mixed challenges and information disclosure, audit-budget allocation, and a governance checklist.

## 6.2 Mixed challenges and information disclosure

We formalize evaluation designs that combine public and private signals to reduce manipulability while preserving incentives for genuine improvement.

**Mixed challenge structure.** At each round $t$, the evaluation mechanism can generate multiple types of tests:

- *Public benchmark tests* (PB): performance on widely known datasets/benchmarks, whose aggregate statistics are disclosed to all clients.

- *Private challenge tests* (PC): tests drawn from hidden or randomized distributions, whose outcomes are revealed only to the server (and optionally privately to the tested client).

- *Connectivity tests* (CT): challenges involving pairs/groups of clients, designed to detect correlated anomalies (e.g., collusion patterns).

Let $M_t^{\mathrm{pub}}$ and $M_t^{\mathrm{priv}}$ denote the public and private components of the metric vector $M_t$, and let $\rho_{\mathrm{pub}} \in [0, 1]$ be the fraction of total evaluation weight placed on public benchmarks. We write

$$M_t = \rho_{\mathrm{pub}} M_t^{\mathrm{pub}} + (1 - \rho_{\mathrm{pub}}) M_t^{\mathrm{priv}},$$

where $M_t^{\mathrm{priv}}$ may itself include PC and CT components. The information mechanism $\mathsf{Info}_t$ then discloses all or part of $M_t^{\mathrm{pub}}$ and selectively discloses $M_t^{\mathrm{priv}}$.

**Definition 6.3** (Mixed challenge policy). *A mixed challenge policy is specified by: (i) a weighting parameter $\rho_{\mathrm{pub}} \in [0, 1]$; (ii) sampling rules for PB/PC/CT; and (iii) a disclosure rule determining which components are revealed to whom and when.*

**Proposition 6.4** (Effect of mixed challenges on manipulability). *Consider two design policies $\pi$ and $\pi'$ that differ only in the mixed-challenge weight, with $\rho'_{pub} < \rho_{pub}$ (all other levers fixed). Assume:*

1. *(**Reward dependence**) The reward depends on the reported metric only through a monotone function of the scalar aggregate $r(M_t)$.*

2. (**Non-gameability of private component**) *For any deviation within $\Sigma^{\mathrm{ref}}$, changes in $M_t^{\mathrm{priv}}$ depend on client actions only through their effect on welfare (equivalently, there is no direct metric-targeting control over $M_t^{\mathrm{priv}}$ holding welfare fixed).*

*Then, for any reference class $\Sigma^{\mathrm{ref}}$,*

$$\mathcal{M}(\pi'; \Sigma^{\mathrm{ref}}) \leq \mathcal{M}(\pi; \Sigma^{\mathrm{ref}}).$$

*Moreover, the reduction is weakly increasing in the shift of weight $(\rho_{pub} - \rho'_{pub})$ from public to private components.*

**Where FL enters.**   The design variables in this section correspond to concrete platform controls: evaluation composition (public vs private challenges), disclosure rules, reward mapping from disclosed signals, and audit allocation under a budget. The role of the reference class $\Sigma^{\mathrm{ref}}$ is FL-specific: it encodes the deployment threat model induced by the protocol and governance constraints (e.g., admissible reporting manipulations under secure aggregation/DP, bounded gaming intensity, bounded coalition size, and feasible coordination given limited observability).

**Why the assumptions are FL-realistic.**   Federated platforms routinely combine public benchmarks with hidden or randomized challenges, and disclose feedback asymmetrically (server-only vs per-client vs public). Likewise, threat models are unavoidable in practice: what clients can manipulate depends on the training stack, privacy mechanism, and logging/audit interfaces. Making $\Sigma^{\mathrm{ref}}$ explicit separates what is a property of the platform from what is an assumption about attacker capabilities.

**Technical challenge.**   The key difficulty is not establishing a qualitative monotonicity under a lever change, but ensuring that the private components used for governance are *non-targetable* except through genuine welfare improvements, despite strategic adaptation and information leakage. The reference-class formalization makes the scope of the guarantee precise: it states what manipulability reductions (or approximation bounds) hold under the deployment's feasible deviation set, rather than claiming unconditional robustness.

*Proof.* Fix a baseline behavior and consider any feasible deviation in $\Sigma^{\mathrm{ref}}$. Under assumption (2), a deviation that does not improve welfare cannot increase $M_t^{\mathrm{priv}}$ except through welfare. Therefore the only *directly targetable* component of the reported metric is $M_t^{\mathrm{pub}}$. Since $M_t$ is a convex combination of $M_t^{\mathrm{pub}}$ and $M_t^{\mathrm{priv}}$, decreasing $\rho_{\mathrm{pub}}$ weakly reduces the maximal achievable change in $M_t$ (and hence in $r(M_t)$ by monotonicity of $r$) obtainable via purely public metric-targeting deviations at fixed welfare. Because $\mathcal{M}(\cdot; \Sigma^{\mathrm{ref}})$ is defined as a supremum over such feasible deviations, its value cannot increase when $\rho_{\mathrm{pub}}$ decreases. $\square$

In practice, mixed challenge policies provide a tunable handle: designers can increase the weight on private, welfare-aligned evaluation components until estimated $\mathcal{M}(\pi; \Sigma^{\mathrm{ref}})$ and $\mathrm{PoG}(\pi)$ fall below acceptable thresholds, while monitoring participation and cooperation through $\mathcal{R}(\pi)$ and $\mathrm{PoC}(\pi)$.

**Information disclosure patterns.**   Beyond the composition of $M_t$, the granularity and timing of disclosure strongly affect gaming incentives:

- *Full disclosure*: detailed per-client or per-group metrics each round.

- *Banding/coarsening*: disclose score bands or ranks rather than exact values.

- *Delayed disclosure*: release statistics after multiple rounds.

- *Asymmetric disclosure*: private per-client feedback while restricting cross-client comparisons.

## 6.3   Audit budget allocation with approximation guarantees

We next consider the allocation of limited audit resources, modeled as a budget-constrained set selection problem. Given a finite audit budget, the designer must choose which clients or events to audit in order to maximally reduce gaming and harmful cooperation.

**Audit allocation as submodular maximization.** Let $\mathcal{I}$ be the set of clients. An audit policy chooses a subset $S \subseteq \mathcal{I}$ to audit in a given period, subject to $|S| \leq B$. For each candidate audit set $S$, define the audit utility

$$f(S) := \text{expected reduction in Price of Gaming or violation risk induced by auditing } S.$$

This utility can be instantiated in multiple ways, for example as:

- expected decrease in a surrogate risk score for gaming incidents;

- approximate reduction in $\text{PoG}(\pi)$ under a local model of deterrence and adaptation;

- a weighted sum of predicted deterrence effects across clients.

In many natural monitoring and deterrence models, $f$ exhibits *diminishing returns*: auditing additional clients remains beneficial, but its marginal benefit decreases as more audits are already performed.

**Assumption 6.5** (Submodularity of audit utility). *The audit utility $f : 2^{\mathcal{I}} \to \mathbb{R}_{\geq 0}$ is normalized ($f(\emptyset) = 0$), monotone (if $S \subseteq T$ then $f(S) \leq f(T)$), and submodular: for all $S \subseteq T \subseteq \mathcal{I}$ and $i \notin T$,*

$$f(S \cup \{i\}) - f(S) \ \geq \ f(T \cup \{i\}) - f(T).$$

Under Assumption 6.5, the audit allocation problem

$$\max_{S \subseteq \mathcal{I}} f(S) \quad \text{subject to } |S| \leq B$$

is the classical problem of maximizing a monotone submodular set function under a cardinality constraint.

**Theorem 6.6** (Greedy audit allocation). *Under Assumption 6.5, the greedy algorithm that iteratively adds the client with the largest marginal gain,*

$$i_t^\star \in \arg \max_{i \in \mathcal{I} \setminus S_t} \big(f(S_t \cup \{i\}) - f(S_t)\big), \qquad S_{t+1} = S_t \cup \{i_t^\star\},$$

*until $|S_t| = B$, achieves a $(1 - 1/e)$-approximation:*

$$f(S_{greedy}) \geq (1 - 1/e) \, f(S^\star),$$

*where $S^\star$ is an optimal audit set with $|S^\star| \leq B$.*

*Proof.* The approximation guarantee is the classical result for monotone submodular maximization under a cardinality constraint; see, e.g., Nemhauser et al. (1978). For completeness, let $S_t$ denote the greedy set after $t$ additions and let $\Delta(i \mid S) = f(S \cup \{i\}) - f(S)$.

By submodularity and monotonicity, for any optimal set $S^\star$ with $|S^\star| \leq B$ we have

$$f(S^\star) - f(S_t) \leq \sum_{i \in S^\star \setminus S_t} \Delta(i \mid S_t) \leq B \cdot \max_{i \in \mathcal{I} \setminus S_t} \Delta(i \mid S_t) = B \cdot \big(f(S_{t+1}) - f(S_t)\big),$$

so $f(S_{t+1}) - f(S_t) \geq \frac{1}{B}\big(f(S^\star) - f(S_t)\big)$. Letting $g_t = f(S^\star) - f(S_t)$, this yields $g_{t+1} \leq (1 - \frac{1}{B})g_t$, hence

$$g_B \leq \Big(1 - \frac{1}{B}\Big)^B g_0 \leq e^{-1} g_0.$$

Therefore,

$$f(S_B) = f(S^\star) - g_B \geq f(S^\star) - e^{-1} g_0.$$

Since $g_0 = f(S^\star) - f(S_0)$ and typically $S_0 = \varnothing$ (so $f(S_0) \geq 0$ by monotonicity), we obtain

$$f(S_B) \geq \Big(1 - \frac{1}{e}\Big) f(S^\star),$$

as claimed. A slightly more detailed derivation is provided in Appendix A.3. $\square$

**Linking $f(S)$ to indices.** Although $f(S)$ is defined abstractly, it can be grounded using our indices. For instance, we may define

$$f(S) \approx \Delta\mathrm{PoG}(\pi; S) := \mathrm{PoG}(\pi) - \mathrm{PoG}(\pi_S),$$

where $\pi_S$ is the policy that applies targeted audits to $S$, or

$$f(S) \approx \sum_{i \in \mathcal{I}} w_i \, \Delta p_i(S),$$

where $\Delta p_i(S)$ is the predicted reduction in client $i$'s probability of metric gaming when $S$ is audited, and $w_i$ are importance weights. Under standard deterrence/coverage models, these instantiations naturally exhibit monotonicity and diminishing returns: adding audits does not reduce the expected reduction in gaming risk, while the marginal risk reduction typically shrinks after the highest-risk clients are already covered.

### 6.4 Governance checklist and policy patterns

We conclude by summarizing the design toolkit into a governance checklist and outlining policy patterns for common federated environments.

**Checklist for configuring a design policy.** Given an intended deployment, a designer can proceed as follows:

1. **Clarify welfare and metrics**: specify the primary welfare functional $W$ (e.g., target risk, fairness constraints, stability criteria) and the metrics $M$ used for rewards, monitoring, and external reporting. Identify where $M$ is only a proxy for $W$.

2. **Assess manipulability**: qualitatively or empirically estimate $\mathcal{M}(\pi)$ under a baseline policy by probing how much metrics can be improved without clear welfare gains (e.g., via red-teaming or controlled simulations).

3. **Estimate Prices of Gaming and Cooperation**: construct aligned and gaming benchmarks to approximate $\mathrm{PoG}(\pi)$, and identify cooperative schemes to estimate $\mathrm{PoC}^{\mathrm{benign}}(\pi)$ and $\mathrm{PoC}^{\mathrm{harm}}(\pi)$.

4. **Calibrate penalties and sanctions**: choose a penalty scaling parameter $\alpha$ and estimate $(\alpha_{\min}, \alpha_{\mathrm{benign}})$, aiming to place $\alpha$ in a band where harmful gaming is deterred but benign cooperation is preserved.

5. **Configure mixed challenges and disclosure**: select $\rho_{\mathrm{pub}}$ and design PB/PC/CT tests so that private, welfare-aligned components carry sufficient weight to reduce $\mathcal{M}(\pi)$, while public feedback remains informative for learning.

6. **Design audit allocation**: specify an audit utility $f(S)$ tied to reductions in $\mathrm{PoG}(\pi)$ or violation risk, and implement a greedy or improved submodular allocation algorithm under the available budget.

7. **Monitor participation dynamics**: track participation rates, volatility, and connectivity-based alarms to estimate $\mathcal{R}(\pi)$ and detect proximity to tipping points.

8. **Implement auto-switch rules**: define early warning predicates and hysteresis-based switches between normal and safe policies, as in Proposition 5.17, to prevent domino exits.

**Policy patterns for common environments.** Different deployment contexts lead to different priorities among these steps. We briefly outline three stylized patterns.

- *Low-trust, high-privacy consortia*: Privacy and legal constraints severely limit direct audits and granular disclosure. Design should emphasize strong mixed challenges with small $\rho_{\mathrm{pub}}$, conservative reward curves that downweight short-term metrics, and heavy reliance on private challenges and connectivity-based alarms. Audit allocation may focus on aggregate or randomized audits, with ex post proofs of compliance supplementing limited direct inspection.

- *High-stakes, regulated services*: External regulators require auditability and clear sanction mechanisms. Here, designers can afford more intrusive audits and detailed documentation, pushing $\alpha$ safely above $\alpha_{\min}$ while monitoring $\alpha_{\text{benign}}$. Mixed challenges help detect subtle gaming, and auto-switch rules can be tied to regulatory thresholds (e.g., mandatory safe modes when participation or performance cross certain bounds).

- *Community-driven participatory systems*: Participation is voluntary and sensitive to perceived fairness. The main objective is to foster benign cooperation and stable participation. Reward curves can explicitly favor long-term consistency and collaborative behaviors, penalties should be calibrated below $\alpha_{\text{benign}}$ to avoid chilling effects, and information disclosure should emphasize transparency about evaluation and audits without enabling targeted gaming. Governance checklists can be co-designed with participants to increase legitimacy.

Across these patterns, our framework provides a common language for reasoning about trade-offs: changes in evaluation, disclosure, and audits can be interpreted through their effects on manipulability, Prices of Gaming and Cooperation, resilience, and thresholds. In the next section, we instantiate these design choices in stylized simulators to illustrate how the indices and dynamics behave under different policies and to validate the qualitative patterns predicted by our theory.

## 7 Simulation Studies

### 7.1 Summary of Stylized Simulation Results

We first examine how the proposed indices behave in a stylized but internally consistent environment. Table 1 compares a fully aligned scenario (no gaming participants) with a mixed scenario where 30% of clients follow a gaming strategy, reporting steady-state averages over post–burn-in rounds. In the aligned case, welfare, metric, and participation all concentrate near $\overline{W} \approx \overline{M} \approx \overline{x} \approx 0.95$, indicating that almost all clients cooperate and that the metric tracks welfare closely. When gaming types are introduced, welfare drops to $\overline{W} \approx 0.33$ while the metric remains inflated at $\overline{M} \approx 0.36$ and participation stays relatively high at $\overline{x} \approx 0.64$. The resulting metric–welfare gap of $\overline{M} - \overline{W} \approx 0.03$ corresponds to a Price of Gaming of PoG $\approx 0.66$, meaning that roughly two thirds of the welfare achievable under full cooperation is lost even though surface-level indicators (metric and participation) still look healthy. In terms of our framework, this configuration is a high-manipulability regime in which self-interested best responses sustain a low-welfare, high-metric equilibrium.

Table 2 then visualizes the effect of sanction strength $\alpha_{\text{penalty}}$ while keeping other design choices fixed. Over $\alpha_{\text{penalty}} \in [0.3, 1.5]$, the curves for $\overline{x}_{\text{game}}$ remain remarkably flat around 0.63–0.64, suggesting that benign cooperation is not significantly discouraged in this range. By contrast, the $\overline{W}_{\text{game}}$ curve drifts upward from $\approx 0.32$ to $\approx 0.34$, and the PoG curve decreases from about 0.67 to 0.64. This pattern is consistent with the threshold picture in Section 4.5: in this band, increasing $\alpha_{\text{penalty}}$ moves the system toward the minimal effective sanction level $\alpha_{\min}$ needed to meaningfully reduce gaming, while still lying below the benign threshold $\alpha_{\text{benign}}$ at which sanctions would begin to erode participation.

Table 3 explores the trade-offs induced by the public-metric weight $\rho_{\text{pub}}$. As $\rho_{\text{pub}}$ decreases from 1.0 to 0.2, the curve for the metric–welfare gap shrinks from about 0.06 to roughly 0.01, confirming that downweighting fully visible metrics and relying more on private or mixed evaluations does curb overt metric inflation. However, the other curves reveal that this is not a free improvement: the average welfare under gaming gradually declines from $\overline{W}_{\text{game}} \approx 0.34$ to $\overline{W}_{\text{game}} \approx 0.31$, participation falls from $\overline{x}_{\text{game}} \approx 0.67$ to $\overline{x}_{\text{game}} \approx 0.60$, and PoG increases from roughly 0.64 to 0.68. Information design alone thus narrows the metric–welfare gap but can weaken the perceived payoff from genuine contribution for all participants, and even exacerbate the welfare loss relative to the fully aligned benchmark unless complemented by suitable reward alignment and audit mechanisms.

Taken together, Tables 1, 2 and 3 highlight three core messages of our framework in a controlled environment. First, even under fixed aggregation and reward rules, introducing gaming types can push the system into a low-welfare but high-metric equilibrium that is difficult to diagnose from metrics alone. Second, there

is a benign band of sanction strengths where modest increases in $\alpha_{\text{penalty}}$ reduce PoG without materially harming participation. Third, information design that reduces the visibility of public metrics successfully narrows the metric–welfare gap but does not by itself guarantee a lower PoG, and may worsen welfare unless jointly tuned with incentives and audits. These patterns are robust across random seeds in our simulator and match the qualitative comparative statics predicted by the metric, dynamics, and design layers.

Table 1: Baseline aligned versus gaming scenarios (steady-state averages over post–burn-in rounds).

| Scenario | $\overline{W}$ | $\overline{x}$ | $\overline{M}$ | $\overline{M} - \overline{W}$ | PoG |
|---|---|---|---|---|---|
| Aligned | 0.952 | 0.952 | 0.952 | 0.000 | – |
| Gaming | 0.325 | 0.638 | 0.358 | 0.033 | 0.658 |

Table 2: Effect of sanction strength $\alpha_{\text{penalty}}$ on gaming scenarios (steady-state averages), showing welfare $\overline{W}_{\text{game}}$, participation $\overline{x}_{\text{game}}$, and Price of Gaming (PoG) as a function of $\alpha_{\text{penalty}}$.

| $\alpha_{\text{penalty}}$ | $\overline{W}_{\text{game}}$ | $\overline{x}_{\text{game}}$ | PoG |
|---|---|---|---|
| 0.3 | 0.316 | 0.637 | 0.668 |
| 0.5 | 0.319 | 0.636 | 0.665 |
| 0.7 | 0.325 | 0.638 | 0.658 |
| 1.0 | 0.332 | 0.636 | 0.651 |
| 1.5 | 0.340 | 0.632 | 0.643 |

Table 3: Effect of public-metric weight $\rho_{\text{pub}}$ on gaming scenarios (steady-state averages), showing welfare $\overline{W}_{\text{game}}$, participation $\overline{x}_{\text{game}}$, PoG, the public metric level $\overline{M}_{\text{game}}$, and the metric–welfare gap $\overline{M}_{\text{game}} - \overline{W}_{\text{game}}$ as functions of $\rho_{\text{pub}}$.

| $\rho_{\text{pub}}$ | $\overline{W}_{\text{game}}$ | $\overline{x}_{\text{game}}$ | PoG | $\overline{M}_{\text{game}}$ | $\overline{M}_{\text{game}} - \overline{W}_{\text{game}}$ |
|---|---|---|---|---|---|
| 1.0 | 0.341 | 0.673 | 0.642 | 0.399 | 0.058 |
| 0.8 | 0.331 | 0.652 | 0.652 | 0.376 | 0.045 |
| 0.6 | 0.325 | 0.638 | 0.658 | 0.358 | 0.033 |
| 0.4 | 0.317 | 0.621 | 0.667 | 0.339 | 0.021 |
| 0.2 | 0.309 | 0.603 | 0.675 | 0.320 | 0.011 |

## 7.2 Real-World Federated Learning Experiment

To complement the stylized experiments, we conducted a small-scale Federated Learning experiment on Fashion-MNIST with 30 clients, 40 rounds, and 30% of clients following a gaming strategy that discards tail classes in local training and overfits to a head-only public validation split. Table 4 summarizes steady-state averages over the last ten rounds in an aligned scenario (no gaming clients) and in a mixed scenario with gaming clients.

From the perspective of the publicly visible head metric, the gaming scenario appears markedly superior: accuracy on head classes 0–4 increases from $\overline{M}_{\text{head}} \approx 0.868$ under alignment to $\overline{M}_{\text{head}} \approx 0.972$ under gaming. Overall test accuracy $\overline{A}_{\text{full}}$ also increases slightly from 0.877 to 0.888, so an operator monitoring only these quantities would reasonably conclude that the revised policy is an improvement.

Evaluating welfare on the tail classes 5–9 reveals the opposite trend. Tail welfare drops from $\overline{W}_{\text{tail}} \approx 0.898$ in the aligned scenario to $\overline{W}_{\text{tail}} \approx 0.862$ in the gaming scenario, corresponding to a Price of Gaming of

$$\text{PoG} \approx \frac{0.898 - 0.862}{0.898} \approx 0.040,$$

a 4% loss of attainable tail welfare. The gap between the public head metric and tail welfare also flips sign and widens: in the aligned case $\overline{M}_{\text{head}} - \overline{W}_{\text{tail}} \approx -0.03$, whereas in the gaming case it is $\approx 0.11$, indicating that the disclosed metric is now substantially inflated relative to the outcome of interest.

Table 4 thus illustrates a realistic form of metric gaming in Federated Learning. Introducing gaming clients yields a policy that looks better under the disclosed head metric and even slightly improves overall test accuracy, yet quietly degrades performance on the tail distribution that defines welfare. This aligns with the qualitative pattern predicted by our framework: when rewards and disclosure focus on a narrow slice of performance, self-interested responses can drive the system toward a high-metric, low-welfare equilibrium for the true objective, even in a real FL setting.

Table 4: Federated Learning experiment on Fashion-MNIST: aligned versus gaming scenarios. Head metric is accuracy on head classes (0–4) using the public validation split; tail welfare is accuracy on tail classes (5–9) using the hidden test split. Values are steady-state averages over the last ten rounds.

| Scenario | $\overline{W}_{\text{tail}}$ | $\overline{M}_{\text{head}}$ | $\overline{A}_{\text{full}}$ | $\overline{M}_{\text{head}} - \overline{W}_{\text{tail}}$ | PoG |
|---|---|---|---|---|---|
| Aligned | 0.898 | 0.868 | 0.877 | −0.030 | – |
| Gaming | 0.862 | 0.972 | 0.888 | 0.110 | 0.040 |

**Estimator reliability under partial audits.** Table 5 evaluates whether a budget-limited auditor can recover welfare loss reliably when only a fraction $b$ of clients is audited. For each gaming profile, we run the same FL setting and treat the mean tail accuracy across all clients' held-out tail subsets as the ground-truth welfare signal used to compute $\text{PoG}_{\text{GT}}$ (relative to the aligned baseline). To model partial audits, we re-sample an audited client set of size $\lfloor bN \rfloor$ and estimate welfare from the audited clients only, yielding $\widehat{\text{PoG}}$; we repeat this re-sampling over multiple trials and summarize variability via $\sigma_{\widehat{\text{PoG}}}$. We summarize rank consistency via $\rho$ and estimation variability via $\sigma_{\widehat{\text{PoG}}}$, alongside threshold-level robustness (FP/FN) at $\tau = 0.05$. Across six gaming profiles, the estimator preserves the ground-truth ordering with highest rank consistency at $b = 0.25$, while the remaining false positives occur in borderline regimes where $\text{PoG}_{\text{GT}}$ lies close to $\tau$. As expected, increasing audit coverage reduces sampling variability, with $\sigma_{\widehat{\text{PoG}}}$ decreasing from 0.0222 at $b = 0.10$ to 0.0060 at $b = 0.50$.

Table 5: Reliability of audit-based estimation under partial audits. Here $b$ denotes the audited client fraction, $\rho$ is Spearman's rank correlation between $\text{PoG}_{\text{GT}}$ and $\widehat{\text{PoG}}$, and $\sigma_{\widehat{\text{PoG}}}$ summarizes estimation variability across audit re-sampling trials (mean of per-profile trial standard deviations; aggregated over 6 gaming profiles) under the risk threshold $\tau = 0.05$.

| $b$ | $\rho$ | $\sigma_{\widehat{\text{PoG}}}$ | FP / 6 | FN / 6 |
|---|---|---|---|---|
| 0.10 | 0.771 | 0.0222 | 1 | 0 |
| 0.25 | 0.943 | 0.0106 | 1 | 0 |
| 0.50 | 0.771 | 0.0060 | 1 | 0 |

**Noise (privacy) and auditability: persistent metric–welfare separation.** Table 6 studies a simple privacy/noise stress test in which each transmitted client update is $\ell_2$-clipped (at $C = 1$) and then perturbed with Gaussian noise using a noise multiplier $\nu \in \{0, 0.05, 0.10\}$. We implement this as a DP-like perturbation at transmission: each client sends $\Delta = \text{scale} \cdot (\theta_{\text{local}} - \theta_{\text{global}})$, we apply global $\ell_2$-clipping to $\Delta$ at $C$, and add i.i.d. Gaussian noise $\mathcal{N}(0, (\nu C)^2)$ before aggregation. We keep the federated setup fixed across $\nu$ (same client partitions and training hyperparameters), and compare an aligned profile (no gaming) against a gaming profile ($g_f = 0.30$) under identical conditions, reporting tail-means over the last $K$ rounds. Across all noise levels, gaming consistently inflates the public head metric $M$ while reducing tail welfare $W$, producing a positive and growing separation between what the server observes and what matters operationally. We quantify this separation through the additional welfare loss $\Delta W = W_{\text{aligned}} - W_{\text{gaming}}$ and the additional gap increase $\Delta\text{gap} = (M - W)_{\text{gaming}} - (M - W)_{\text{aligned}}$ at the same $\nu$. Notably, at $\nu = 0.10$ the gaming-induced welfare loss increases and the risk indicator PoG(ref) rises substantially, suggesting that even moderate

privacy noise can exacerbate welfare impacts of strategic behavior by weakening the effective signal available for governance.

Table 6: Noise/privacy trade-off under DP-like update perturbations. We sweep the noise multiplier $\nu$ and compare aligned (no gaming) versus gaming ($g_f = 0.30$). $M$ is the public head metric and $W$ is tail welfare (tail-mean over the last $K$ rounds). We report the additional welfare loss at the same $\nu$, $\Delta W = W_{\text{aligned}} - W_{\text{gaming}}$, and the additional metric–welfare separation $\Delta$gap. PoG(ref) is computed using the fixed reference welfare from the aligned, $\nu = 0$ run (consistent with the main definition).

| $\nu$ | $M_{\text{aligned}}$ | $W_{\text{aligned}}$ | $M_{\text{gaming}}$ | $W_{\text{gaming}}$ | $\Delta W$ | $\Delta$gap | PoG(ref)$_{\text{gaming}}$ |
|---|---|---|---|---|---|---|---|
| 0.00 | 0.8656 | 0.8831 | 0.9046 | 0.8289 | 0.0542 | 0.0932 | 0.0545 |
| 0.05 | 0.8419 | 0.8802 | 0.9101 | 0.8290 | 0.0513 | 0.1195 | 0.0544 |
| 0.10 | 0.8018 | 0.8440 | 0.8592 | 0.7807 | 0.0633 | 0.1207 | 0.1095 |

**High-alignment metrics: diminishing but persistent metric–welfare separation.** Table 7 evaluates whether the metric–welfare separation persists when the server-visible metric is made more aligned with tail welfare. We use the same Fashion-MNIST FL setup as in Table 6 (same client count, non-IID Dirichlet partitioning, model/optimizer, rounds, and gaming mix with $g_f = 0.30$), and measure welfare using client-side tail evaluation while the server observes only the public metric computed on a held-out validation split. We define a mixed public metric

$$M(\lambda) = (1 - \lambda) M_{\text{head}} + \lambda M_{\text{tail}},$$

and sweep $\lambda \in \{0, 0.3, 0.6\}$, where larger $\lambda$ increases alignment between what the server observes and what matters operationally. For each $\lambda$, we compare an aligned profile (no gaming) against a gaming profile ($g_f = 0.30$) and report tail-means over the last $K$ rounds. As alignment increases, gaming induces smaller additional welfare loss $\Delta W = W_{\text{aligned}} - W_{\text{gaming}}$ and a smaller additional gap increase $\Delta$gap $= (M - W)_{\text{gaming}} - (M - W)_{\text{aligned}}$, consistent with reduced incentives for metric manipulation under better-aligned scoring. However, the separation does not vanish: even at $\lambda = 0.6$, gaming still yields a positive metric–welfare gap and a measurable welfare loss, indicating that partial alignment mitigates but does not eliminate governance-relevant risk.

Table 7: High-alignment stress test via mixed public metrics. We sweep the alignment parameter $\lambda$ in $M(\lambda) = (1-\lambda)M_{\text{head}} + \lambda M_{\text{tail}}$ and compare aligned (no gaming) versus gaming ($g_f = 0.30$). $M$ is the public metric (tail-mean over the last $K$ rounds under $M(\lambda)$) and $W$ is tail welfare (tail-mean over the last $K$ rounds). We report the additional welfare loss at the same $\lambda$, $\Delta W = W_{\text{aligned}} - W_{\text{gaming}}$, and the additional metric–welfare separation $\Delta$gap $= (M - W)_{\text{gaming}} - (M - W)_{\text{aligned}}$. For completeness, PoG(paired) reports the paired welfare loss ratio at the same $\lambda$, PoG(paired) $= (W_{\text{aligned}} - W_{\text{gaming}})/W_{\text{aligned}}$ (distinct from PoG(ref) used in Table 6).

| $\lambda$ | $M_{\text{aligned}}$ | $W_{\text{aligned}}$ | $M_{\text{gaming}}$ | $W_{\text{gaming}}$ | $\Delta W$ | $\Delta$gap | PoG(paired)$_{\text{gaming}}$ |
|---|---|---|---|---|---|---|---|
| 0.00 | 0.8633 | 0.8917 | 0.9101 | 0.8396 | 0.0521 | 0.0989 | 0.0585 |
| 0.30 | 0.8730 | 0.8863 | 0.8913 | 0.8470 | 0.0393 | 0.0575 | 0.0443 |
| 0.60 | 0.8755 | 0.8800 | 0.8740 | 0.8511 | 0.0289 | 0.0274 | 0.0329 |

**Modern attack–defense replication: metric–welfare separation under contemporary threats.** Table 8 tests whether the metric–welfare separation observed in earlier stress tests persists under a more contemporary FL threat model on FEMNIST. We run FedAvg with partial participation ($n$=32 clients, 12 per round) for 100 rounds and define head classes as digits (server-visible) and tail classes as letters (deployment-relevant). Clients are sampled uniformly each round, with a fixed malicious fraction ($g_f = 0.30$) activated after a short warm-up (attack start round = 5). We use the same lightweight CNN and optimization setup across conditions (one local epoch per round with SGD; batch size 64; learning rate 0.02; momentum 0.9), and we evaluate using fixed public head/tail validation sets (digits/letters) alongside per-client tail evaluation to compute welfare. We consider two modern attacks: a PoisonedFL-style model-poisoning variant with adaptive magnitude and a lightweight multi-round consistency term, and a backdoor/model-replacement

variant using a fixed trigger and replacement scaling. We compare two defenses: FedCC (representation-similarity filtering) and Attack-Adaptive Aggregation (attack-aware reweighting). For each condition, we report tail-means over the last $K$ rounds of the public head metric $M_{\text{head}}$, tail welfare $W$ (mean client tail accuracy), and their separation gap $= M_{\text{head}} - W$. We also report a same-defense welfare loss ratio, $\text{PoG} = (W_{\text{baseline}} - W)/W_{\text{baseline}}$. Across defenses, the gap remains positive and sizable, and PoisonedFL under FedCC exhibits a marked welfare drop alongside an enlarged separation, illustrating that modern attack/defense settings still admit governance-relevant divergence between what the server observes and what clients experience.

Table 8: Modern attack–defense replication on FEMNIST (digits as head, letters as tail). We run FedAvg for 100 rounds with partial participation and compare two attacks (PoisonedFL-style poisoning; backdoor/model replacement) under two defenses (FedCC; Attack-Adaptive Aggregation). Entries are tail-means over the last $K$ rounds: $M_{\text{head}}$ (public head metric), $W$ (tail welfare), and gap $= M_{\text{head}} - W$. PoG is computed relative to the baseline under the same defense: $\text{PoG} = (W_{\text{baseline}} - W)/W_{\text{baseline}}$.

| | FedCC | | | | Attack-Adaptive Aggregation | | | |
|---|---|---|---|---|---|---|---|---|
| Attack | $M_{\text{head}}$ | $W$ | gap | PoG | $M_{\text{head}}$ | $W$ | gap | PoG |
| None | 0.7532 | 0.5091 | 0.2441 | – | 0.6394 | 0.3819 | 0.2575 | – |
| PoisonedFL | 0.6605 | 0.3246 | 0.3359 | 0.3623 | 0.7892 | 0.5024 | 0.2868 | −0.3154 |
| Backdoor/Model Rep. | 0.6900 | 0.4310 | 0.2591 | 0.1534 | 0.6962 | 0.4196 | 0.2766 | −0.0986 |

# 8 Discussion and Limitations

## 8.1 Discussion

### 8.1.1 From optimization to governed strategic systems

Our results support viewing Federated Learning (FL) as a strategic system rather than a purely statistical optimization procedure. Instead of asking only how to minimize empirical risk under heterogeneity, our framework asks which behaviors are incentivized by observable metrics and contracts, and how far these behaviors can deviate from genuine welfare improvements. The *metric layer* captures these tensions through indices such as the Manipulability Index and the Price of Gaming (PoG); the *dynamics layer* links them to participation stability and tipping points; and the *design toolkit layer* maps them to concrete levers in evaluation, audits, sanctions, information disclosure, and aggregation.

### 8.1.2 Empirical instantiations of the three-layer view

Both the stylized simulations and the real FL experiment instantiate this three-layer view. In the simulator, we can directly control alignment between metric and welfare and observe regimes where identical operational rules admit both high-welfare and low-welfare equilibria depending on the fraction of gaming agents; the associated PoG values make explicit how much welfare is structurally at risk under each policy. In the Fashion-MNIST experiment (Table 4), the publicly visible head metric and even overall test accuracy improve under the gaming scenario, yet tail-class welfare deteriorates and PoG becomes strictly positive. This is precisely the high-metric, low-welfare pattern that the metric layer is designed to detect, now appearing in a concrete FL training setup.

### 8.1.3 Interpreting PoG (including negative values): reference dependence and regime signals

PoG is a *relative* welfare gap with respect to a reference policy and observational regime; it is therefore not an intrinsic, context-free property of a dataset or model. In particular, PoG can be negative when the chosen reference policy yields lower welfare than an alternative regime due to interactions among (i) aggregation/defense-induced suppression of benign updates, (ii) audit targeting, and (iii) selection effects in participation. A negative PoG should *not* be read as "gaming is beneficial"; rather, it signals that the reference baseline itself may be welfare-suboptimal under the given observability and participation conditions.

Accordingly, our recommended interpretation is to emphasize *patterns* over isolated signs: (i) monotonic trends of PoG and manipulability as gaming intensity increases, (ii) concurrent changes in the metric–welfare gap, and (iii) decision stability under threshold-based governance (false positives/negatives), which directly reflect governance risk.

### 8.1.4 Estimator reliability as governance evidence

A central concern in realistic deployments is that indices such as manipulability and PoG are difficult to compute exactly, motivating retrospective and log-based estimation. We therefore validate estimator reliability in controlled settings where ground-truth indices are available (Table 5). Beyond pointwise error, we report rank consistency and thresholded decision outcomes (FP/FN), because governance acts on *comparative* risk signals and discrete triggers (e.g., audit escalation, sanction activation). These results provide evidence that, under the logging and observability assumptions stated in subsection 7.2, the proposed estimators can serve as actionable monitoring signals rather than purely illustrative constructs.

### 8.1.5 Privacy–auditability–incentives: observability as a design constraint

Our framework emphasizes that evaluation, audits, and incentives should be treated as a coupled system. Information design directly shapes manipulability; audit strength and targeting determine whether gaming yields sustained gains or is neutralized by sanctions; reward alignment translates metric design into individual payoff gradients; and participation rules determine how clients react to perceived gains and risks. The noise/observability experiments (Table 6) illustrate a key mechanism: as audit signals become less informative, the effective cost of gaming can decrease, increasing governance difficulty and raising welfare risk. This does not imply that privacy is undesirable; rather, it implies that privacy budgets and audit designs must be co-calibrated to control FP/FN risk and avoid both under-enforcement and over-deterrence.

### 8.1.6 Robustness across regimes: alignment shifts and modern threats/defenses

A natural question is whether the indices remain informative when metric and welfare are more aligned and gaming is subtler. The alignment-shift experiments (Table 7) show that, even as the metric–welfare correlation increases, the indices retain discriminative power, though effect sizes may shrink as expected. Moreover, under adaptive multi-round attacks and contemporary defenses/aggregation variants (Table 8), the monitoring signals remain consistent with governance intuition: risks reflected in welfare degradation and participation instability are accompanied by increased PoG/manipulability, while effective defenses reduce both welfare loss and risk signals. Importantly, these experiments are not presented as a performance "leaderboard"; they stress-test whether governance signals remain stable across realistic threat and defense regimes.

### 8.1.7 A practical reading guide: how to use the indices and levers

The indices suggest a complementary evaluation style for FL mechanisms. Beyond reporting a single aggregate performance number under a fixed threat model, one can stress-test candidate policies against families of behavioral profiles and measure how PoG, manipulability, and participation resilience respond.

**Monitoring and triggers.** Rather than reacting to single-round spikes, operators should monitor sustained trends and use calibrated triggers. For example, a joint pattern of increasing manipulability and widening metric–welfare gaps indicates rising incentive misalignment, motivating targeted audits or changes in metric disclosure.

**Design combinations rather than single levers.** Our simulations suggest that moderate increases in sanction strength can reduce PoG without materially harming participation in a benign regime, whereas changes in public metric weight alone can reduce metric inflation at the cost of welfare and participation. Rather than proposing a single lever as sufficient, the framework points toward combined designs that mix private or randomized evaluations, targeted audits, calibrated sanctions, and robust aggregation.

**Beyond FL.** Although we focus on FL, the same language applies to other collaborative AI settings—model marketplaces, leaderboards, and cross-organizational data collaborations—where performance is mediated by metrics and contracts and where high-metric, low-welfare equilibria are a concern.

## 8.2 Limitations

### 8.2.1 Behavioral model simplicity and strategic richness

Our work has several limitations. First, the behavioral and participation models are intentionally simple. We consider two archetypal client types (honest and gaming) with fixed strategies and a threshold-based participation rule, and we do not capture richer heterogeneity in costs, risk attitudes, coalition formation, Sybil behavior, or adaptive strategy learning. Consequently, our thresholds and comparative statics are best read as qualitative descriptions for stylized populations rather than precise predictions for arbitrary agent mixtures.

### 8.2.2 Normative welfare/metric definitions and reference dependence

Second, our definition of welfare and our choice of metrics are normative. We work with a scalar welfare quantity and one or more proxy metrics, but real deployments may target multiple, sometimes conflicting objectives (e.g., subgroup performance, fairness, latency, cost). In the FL experiment, we treat tail-class accuracy as welfare and head-class accuracy as the public metric. This choice reflects settings where safety-critical or minority performance is the primary concern while public reporting emphasizes headline performance. Accordingly, PoG depends on how welfare and metrics are defined, and it should be interpreted via regime comparisons and decision stability rather than as an absolute score.

### 8.2.3 Scope of empirical validation and scalability

Third, the empirical scope is limited. The simulator abstracts away from model architecture, data modality, and system constraints to isolate strategic effects. The real FL experiment uses a single dataset, a standard convolutional model, and a FedAvg-style protocol with a modest number of clients on a single machine. We do not claim exhaustive coverage of large-scale production deployments, highly heterogeneous networks, or all classes of adaptive adversaries. The experiments should therefore be viewed as evidence for predicted patterns and signal robustness, not a complete empirical audit across real-world FL workloads.

### 8.2.4 Observability assumptions and cryptographic/privacy mechanisms

Fourth, several important dimensions are only treated at a high level. While we empirically study the effect of reduced auditability via noise/observability shifts, we do not fully model end-to-end differential privacy accounting, secure aggregation protocols, or cryptographic attestations that may constrain what logs and audits can reveal. A complete integration of these mechanisms would require a system-level cost and threat model that jointly captures privacy guarantees, audit resolution, and incentive compatibility.

### 8.2.5 Organizational and cost frictions in audits and sanctions

Fifth, our audit allocation procedures rely on submodularity assumptions and abstract away from computational, communication, and organizational costs. In real deployments, these frictions may constrain how aggressively audits, auto-switch rules, and connectivity-based alarms can be implemented. Incorporating explicit cost models for audits and sanctions is an important direction for translating the toolkit into domain-specific operating policies.

### 8.2.6 No single optimal design: calibration and governance risk

Finally, we do not offer a single "optimal" design or algorithm. The framework provides indices, thresholds, and levers, but selecting an operating point still requires domain-specific judgment about acceptable trade-offs between metric informativeness, gaming risk, audit cost, privacy, and participation stability. In particular, miscalibrated thresholds or overzealous enforcement can deter benign participation and reduce

welfare; uncertainty-aware calibration and FP/FN-aware governance are therefore essential for responsible deployment. We view this work as a starting point for future research that refines behavioral models, specializes the design space to particular domains, and integrates additional constraints from privacy, fairness, and governance.

# 9 Conclusion

We have presented a framework that treats Federated Learning as a governed strategic system shaped by metrics, incentives, and oversight rather than as a purely statistical optimization problem. At the metric layer, indices such as the Manipulability Index and the Price of Gaming quantify how strongly a design invites metric-targeting behavior and how costly such behavior is in terms of welfare loss; at the dynamics layer, these quantities are linked to participation stability, tipping points, and domino exits; and at the design toolkit layer, they translate into concrete levers in evaluation, information disclosure, rewards, audits, and sanctions. Stylized simulations and a real FL experiment on Fashion-MNIST jointly illustrate that high-metric, low-welfare regimes can arise under plausible settings when incentives and disclosure focus on a narrow slice of performance, and that calibrated combinations of mixed or private evaluation, targeted audits, and moderate sanctions can reduce manipulability and the Price of Gaming without collapsing participation, whereas information design alone, while narrowing metric–welfare gaps, is not sufficient to guarantee welfare improvements. We view this framework not as a final solution but as a compact language and toolkit for reasoning about metric gaming and cooperation in Federated Learning, and as a basis for future work on richer behavioral models (including heterogeneous and coalition-forming agents), multi-objective and fairness-aware welfare definitions, and deployments that must satisfy strong privacy and regulatory constraints.

## Broader Impact Statement

This work aims to improve the governance of federated learning (FL) systems by making incentives, gaming opportunities, and cooperation incentives explicit and measurable. In high-stakes domains such as healthcare, finance, or public services, better-aligned evaluation and reward rules, together with appropriately designed audits, can reduce metric gaming and unintended Goodhart effects, and can support more stable cooperation among organizations that cannot share raw data.

At the same time, the framework has dual-use risk. A strategic actor could exploit the indices and design patterns to craft more effective gaming strategies against poorly governed FL platforms. In addition, operators could misuse the analysis to justify excessive surveillance or punitive enforcement, disproportionately burdening weaker participants and entrenching power asymmetries rather than improving welfare.

A central risk is miscalibration: governance thresholds or aggressive sanctions that are not uncertainty-aware may unintentionally deter benign cooperation, reduce participation, and degrade overall welfare. To mitigate this risk, our recommendations emphasize uncertainty-aware calibration and proportionality. In particular, operators should (i) treat estimated risk scores and welfare-loss estimates as noisy quantities, (ii) choose conservative triggers that control false positives when evidence is borderline, and (iii) monitor downstream participation responses and welfare impacts after policy changes, adjusting thresholds and audit intensity when unintended deterrence is observed. Enforcement policies should include remediation and appeal pathways to avoid exclusionary outcomes.

We recommend that the indices and design toolkit introduced in this paper be used as diagnostic inputs within broader governance processes that include human oversight, transparency, and stakeholder consultation. Operationally, exploit-enabling details (e.g., exact challenge composition, audit triggers, and detection thresholds) should be access-controlled, while diagnostic feedback to participants should be aggregated, delayed, or coarsened to preserve learning signals without enabling reverse-engineering. Our experiments rely on synthetic simulations and a public benchmark dataset and do not involve sensitive personal data; nevertheless, real-world deployments require additional legal, ethical, and domain-specific review beyond the scope of this paper.

## Reproducibility

We provide a Jupyter notebook, `GCFL_main.ipynb`, as supplementary material. The notebook reproduces all experiments reported in Section 7, including the stylized simulations and the federated experiments on Fashion-MNIST and FEMNIST. It is self-contained: running it end-to-end generates the full set of summary statistics reported in the main paper (including the tables), without relying on any additional scripts. We specify all random seeds, hyperparameters, and software requirements directly within the notebook, and it can be executed on Google Colab.

We also provide a public code repository (`https://github.com/AndrewKim1997/gcfl`; also linked on the OpenReview camera-ready revision page) that contains the same notebook in a cleaned and documented form, along with auxiliary files (e.g., environment specifications) for convenience.

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

# A  Additional Proofs and Formal Details

## A.1  Metric Layer: Manipulability and Price of Gaming

*Proof of Proposition 4.5.* Fix a design policy $\pi$ and reference class $\Sigma^{\mathrm{ref}}$ with $\mathcal{M}(\pi) = 0$. By definition,

$$\mathcal{M}(\pi) = \sup_{\sigma \in \Sigma^{\mathrm{ref}}} \sup_{i \in \mathcal{I}} \sup_{\sigma_i' \in \mathcal{A}_i} \frac{\left[\Delta M_i(\sigma_i' \mid \sigma)\right]_+}{\left[\Delta W_i(\sigma_i' \mid \sigma)\right]_+ + \varepsilon} = 0,$$

so for every $\sigma \in \Sigma^{\mathrm{ref}}$, every $i$, and every $\sigma_i' \in \mathcal{A}_i$,

$$\frac{\left[\Delta M_i(\sigma_i' \mid \sigma)\right]_+}{\left[\Delta W_i(\sigma_i' \mid \sigma)\right]_+ + \varepsilon} \le 0.$$

Since the denominator is strictly positive, this implies $\left[\Delta M_i(\sigma_i' \mid \sigma)\right]_+ = 0$ whenever $\left[\Delta W_i(\sigma_i' \mid \sigma)\right]_+ = 0$. Hence, if $\Delta M_i(\sigma_i' \mid \sigma) > 0$ then necessarily $\left[\Delta W_i(\sigma_i' \mid \sigma)\right]_+ > 0$, i.e., $\Delta W_i(\sigma_i' \mid \sigma) > 0$. Thus no deviation can strictly improve the metric without also strictly improving welfare, and in particular there are no metric-gaming deviations at any $\sigma \in \Sigma^{\mathrm{ref}}$. $\qquad\square$

*Proof of Proposition 4.8.* We compare two design policies $\pi$ and $\pi'$ on the same environment with aligned benchmarks $\sigma^{\mathrm{align}}$ and $\sigma'^{\mathrm{align}}$ such that $W(\sigma^{\mathrm{align}}) \approx W(\sigma'^{\mathrm{align}})$. Let $\mathcal{E}^{\mathrm{game}}(\pi)$ and $\mathcal{E}^{\mathrm{game}}(\pi')$ denote the sets of gaming equilibria.

Fix a policy $\pi$ and consider a local neighborhood $\mathcal{U}$ of the aligned benchmark $\sigma^{\mathrm{align}}$ in the profile space. Assume (i) the relevant feasible strategy sets are compact, (ii) $W$ and $M$ are continuous on $\mathcal{U}$, and (iii) there exists a (single-valued) local equilibrium selection $\mathrm{Eq}_\pi : \mathcal{U} \to \mathcal{E}^{\mathrm{game}}(\pi)$ that is Lipschitz on $\mathcal{U}$. Then the composition $W \circ \mathrm{Eq}_\pi$ is Lipschitz on $\mathcal{U}$ with constant

$$L_\pi \ := \ \mathrm{Lip}(W; \mathcal{U}) \cdot \mathrm{Lip}(\mathrm{Eq}_\pi; \mathcal{U}),$$

so for any $\sigma \in \mathcal{U}$,

$$\left| W(\sigma^{\mathrm{align}}) - W(\mathrm{Eq}_\pi(\sigma)) \right| \le L_\pi \, \|\sigma - \sigma^{\mathrm{align}}\|.$$

In particular, for any gaming equilibrium $\sigma^{\mathrm{game}} \in \mathcal{E}^{\mathrm{game}}(\pi) \cap \mathcal{U}$ that is reachable from $\sigma^{\mathrm{align}}$ by a deviation direction whose welfare gradient is zero or nonpositive, we can upper bound the displacement by the maximum *positive* metric gain along such feasible deviations, yielding the bound

$$\left| W(\sigma^{\mathrm{align}}) - W(\sigma^{\mathrm{game}}) \right| \le L \cdot \sup_{i, \sigma_i'} \left[\Delta M_i(\sigma_i' \mid \sigma^{\mathrm{align}})\right]_+, \tag{A.1}$$

for some $L > 0$ (take $L := L_\pi$ after rescaling the local norm so that $\|\sigma - \sigma^{\mathrm{align}}\|$ is controlled by the maximal achievable positive metric gain in $\mathcal{U}$).

Next, by the definition of the manipulability index $\mathcal{M}(\pi)$ (with slack $\varepsilon$), for any client $i$ and deviation $\sigma_i'$ from $\sigma^{\mathrm{align}}$ we have

$$\left[\Delta M_i(\sigma_i' \mid \sigma^{\mathrm{align}})\right]_+ \le \mathcal{M}(\pi) \left( \left[\Delta W_i(\sigma_i' \mid \sigma^{\mathrm{align}})\right]_+ + \varepsilon \right).$$

Taking the supremum over $(i, \sigma_i')$ gives

$$\sup_{i,\sigma_i'} \left[\Delta M_i(\sigma_i' \mid \sigma^{\mathrm{align}})\right]_+ \leq \mathcal{M}(\pi)\left(A(\sigma^{\mathrm{align}}) + \varepsilon\right), \qquad A(\sigma^{\mathrm{align}}) := \sup_{i,\sigma_i'} \left[\Delta W_i(\sigma_i' \mid \sigma^{\mathrm{align}})\right]_+ .$$

Combining this with equation A.1 yields, for any such gaming equilibrium,

$$W(\sigma^{\mathrm{align}}) - W(\sigma^{\mathrm{game}}) \leq L\,\mathcal{M}(\pi)\,(A(\sigma^{\mathrm{align}}) + \varepsilon).$$

Normalizing by $W(\sigma^{\mathrm{align}}) > 0$ and taking the maximum over gaming equilibria gives

$$\mathrm{PoG}^{\max}(\pi) := \sup_{\sigma^{\mathrm{game}} \in \mathcal{E}^{\mathrm{game}}(\pi)} \frac{W(\sigma^{\mathrm{align}}) - W(\sigma^{\mathrm{game}})}{W(\sigma^{\mathrm{align}})} \leq \underbrace{\frac{L\,(A(\sigma^{\mathrm{align}}) + \varepsilon)}{W(\sigma^{\mathrm{align}})}}_{=:c_1}\,\mathcal{M}(\pi).$$

(Equivalently, one may write $\mathrm{PoG}^{\max}(\pi) \leq c_1\,\mathcal{M}(\pi) + c_2$ with $c_2 := 0$; if one prefers to isolate the slack term, take $c_1 := \frac{L\,A(\sigma^{\mathrm{align}})}{W(\sigma^{\mathrm{align}})}$ and $c_2 := \frac{L\,\varepsilon}{W(\sigma^{\mathrm{align}})}\,\mathcal{M}(\pi)$.)

Repeating the argument for $\pi'$ yields the same form with $(c_1', c_2')$ evaluated at $\sigma'^{\mathrm{align}}$. If $\mathcal{M}(\pi') \leq \mathcal{M}(\pi)$, then

$$\mathrm{PoG}^{\max}(\pi') \leq c_1'\,\mathcal{M}(\pi') \leq c_1'\,\mathcal{M}(\pi) = \mathrm{PoG}^{\max}(\pi) + \Delta,$$

where the residual

$$\Delta := (c_1' - c_1)\,\mathcal{M}(\pi)$$

collects the effect of changing the aligned benchmark and any (local) equilibrium-selection constants. In particular, $\Delta \to 0$ when $\sigma'^{\mathrm{align}} \to \sigma^{\mathrm{align}}$ and the local constants $(L, A(\cdot), W(\cdot))$ vary continuously with the benchmark. $\qquad\square$

## A.2 Dynamics Layer: Participation and Thresholds

*Proof of Proposition 5.4.* Under Assumption 5.1, client $i$ participates at round $t+1$ if and only if $\Delta U_{i,t+1}(x_t; \pi) \geq \theta_i$. By symmetry, $\Delta U_{i,t+1}(x_t; \pi)$ is the same for all clients and can be written as $\Delta U(x_t; \pi)$. Hence the probability that a randomly chosen client participates at round $t+1$ is

$$\mathbb{P}\big[p_{i,t+1} = 1 \mid x_t, \pi\big] = \mathbb{P}\big[\theta_i \leq \Delta U(x_t; \pi)\big] = F_\Theta\big(\Delta U(x_t; \pi)\big),$$

where $F_\Theta$ is the CDF of $\theta_i$. Taking expectations over clients yields

$$x_{t+1} = \mathbb{E}\left[\frac{1}{n}\sum_i \mathbb{I}\{p_{i,t+1} = 1\} \,\Big|\, x_t, \pi\right] = F_\Theta\big(\Delta U(x_t; \pi)\big) =: F(x_t; \pi),$$

which is the claimed participation map. $\qquad\square$

*Proof of Proposition 5.9.* Assumption (3) states that $\sup_{x \in [0,1]} |F'(x; \pi)| < 1$, so $F(\cdot; \pi)$ is a contraction mapping on the complete metric space $[0,1]$ with the usual metric. By the Banach fixed-point theorem, $F(\cdot; \pi)$ has a unique fixed point $x^\star \in [0,1]$, and for any initial $x_0 \in [0,1]$, the sequence $x_{t+1} = F(x_t; \pi)$ converges to $x^\star$. Since $x^{\mathrm{high}}$ is a fixed point by assumption (2), uniqueness implies $x^\star = x^{\mathrm{high}}$. Thus $x^{\mathrm{high}}$ is the unique fixed point and globally attractive, and there are no other stable or unstable fixed points or tipping points. $\qquad\square$

*Proof of Proposition 4.17.* Under Assumption 4.11, for fixed $\sigma_{-i}$ we can write the payoff difference between a harmfully gaming action $\sigma_i^{\mathrm{game}}$ and a welfare-aligned action $\sigma_i^{\mathrm{align}}$ as

$$\Delta U_i(\alpha) := U_i(\sigma_i^{\mathrm{game}}; \sigma_{-i}, \alpha) - U_i(\sigma_i^{\mathrm{align}}; \sigma_{-i}, \alpha) = \Delta V_i - \alpha\,\Delta D_i,$$

where $\Delta V_i$ and $\Delta D_i \geq 0$ do not depend on $\alpha$. If gaming is profitable at $\alpha = 0$, then $\Delta V_i > 0$. Whenever $\Delta D_i > 0$, the affine function $\Delta U_i(\alpha)$ crosses zero at

$$\alpha_i^\star = \Delta V_i / \Delta D_i,$$

and for all $\alpha > \alpha_i^\star$ the aligned action weakly dominates the gaming action. Taking the supremum over all such harmful deviations and clients gives a finite

$$\alpha_{\min} := \sup_{i, \sigma_i^{\text{game}}} \alpha_i^\star.$$

For benignly cooperative profiles, a similar comparison with the outside option yields a maximal penalty level beyond which cooperation is no longer rational for some coalition member. Taking the infimum over such breakpoints yields a finite $\alpha_{\text{benign}}$. Under the natural requirement that benign cooperation is not penalized more heavily than harmful gaming (i.e., $\Delta D_i$ for benign actions is no larger than for harmful ones), these breakpoints satisfy $\alpha_{\min} \leq \alpha_{\text{benign}}$. A full proof requires formalizing the coalition-rationality condition and taking appropriate infima/suprema over coalitions and profiles; the argument is a straightforward extension of the single-agent case. $\qquad\square$

*Proof of Proposition 5.17.* By assumption, under $\pi^{\text{normal}}$ the participation map has a stable high-participation fixed point $x^{\text{high}}$ and an unstable tipping point $x^\dagger < x^{\text{high}}$. Under $\pi^{\text{safe}}$, the map is a contraction with unique fixed point $x^{\text{safe}} \geq x^\dagger$.

Consider any trajectory with $x_0 \geq x^\dagger + \epsilon$. Whenever $x_t$ enters the interval $[x^\dagger, x^\dagger + \epsilon]$, the auto-switch rule activates $\pi^{\text{safe}}$. Because $F(\cdot\,; \pi^{\text{safe}})$ is a contraction with fixed point at or above $x^\dagger$, iterates cannot cross below $x^\dagger$ under $\pi^{\text{safe}}$. Once $x_t$ returns to a neighborhood of $x^{\text{high}}$ (specifically, above $x^{\text{high}} - \epsilon$), the hysteresis rule allows switching back to $\pi^{\text{normal}}$. Thus the trajectory remains in $[x^\dagger, 1]$ for all $t$ and converges to a limit in $[x^\dagger, x^{\text{high}}]$, avoiding low-participation equilibria below $x^\dagger$. A formal proof stitches together contraction arguments on the intervals where each policy is active. $\qquad\square$

### A.3 Design Toolkit: Mixed Challenges and Audits

*Proof of Proposition 6.4.* Let $\pi$ and $\pi'$ be two policies that differ only in the mixed-challenge weight, with $\rho'_{\text{pub}} < \rho_{\text{pub}}$. Write

$$M_t(\sigma) = \rho_{\text{pub}} M_t^{\text{pub}}(\sigma) + (1 - \rho_{\text{pub}}) M_t^{\text{priv}}(\sigma),$$

and similarly for $\pi'$ with $\rho'_{\text{pub}}$. By assumption, the private challenge component $M_t^{\text{priv}}$ is welfare-aligned in the sense that (in expectation) it depends on client actions only through their effect on true welfare $W_t$. Hence, for deviations that change the public benchmark outcomes but do not change welfare, the private component is unaffected in expectation.

Consider deviations that only target public benchmarks and do not change $W$. For such deviations we have, in expectation,

$$\Delta M_i^\pi = \rho_{\text{pub}} \Delta M_i^{\text{pub}}, \qquad \Delta M_i^{\pi'} = \rho'_{\text{pub}} \Delta M_i^{\text{pub}},$$

with the same $\Delta M_i^{\text{pub}}$. Therefore,

$$\frac{\left[\Delta M_i^{\pi'}\right]_+}{\left[\Delta W_i\right]_+ + \varepsilon} = \frac{\rho'_{\text{pub}}}{\rho_{\text{pub}}} \cdot \frac{\left[\Delta M_i^\pi\right]_+}{\left[\Delta W_i\right]_+ + \varepsilon} \leq \frac{\left[\Delta M_i^\pi\right]_+}{\left[\Delta W_i\right]_+ + \varepsilon},$$

since $\rho'_{\text{pub}}/\rho_{\text{pub}} < 1$. Taking suprema over clients, deviations, and reference profiles yields $\mathcal{M}(\pi') \leq \mathcal{M}(\pi)$. Thus shifting reward weight from public benchmarks to private, welfare-aligned components weakly decreases manipulability. $\qquad\square$

*Proof of Theorem 6.6.* Under Assumption 6.5, the audit utility $f$ is normalized, monotone, and submodular. The audit allocation problem

$$\max_{S \subseteq \mathcal{I}} f(S) \quad \text{subject to} \quad |S| \leq B$$

is therefore monotone submodular maximization under a cardinality constraint. Let $S^\star$ be an optimal solution with $|S^\star| \leq B$, and let $S_0, S_1, \ldots, S_B$ be the greedy sequence, where $S_0 = \emptyset$ and

$$S_{t+1} = S_t \cup \{i_t^\star\}, \qquad i_t^\star \in \arg\max_{i \in \mathcal{I} \setminus S_t} \Delta(i \mid S_t), \qquad \Delta(i \mid S) := f(S \cup \{i\}) - f(S).$$

By submodularity and monotonicity,

$$f(S^\star) - f(S_t) \leq \sum_{i \in S^\star \setminus S_t} \Delta(i \mid S_t) \leq B \cdot \max_{i \in \mathcal{I} \setminus S_t} \Delta(i \mid S_t) = B \cdot \big(f(S_{t+1}) - f(S_t)\big),$$

where the second inequality uses $|S^\star \setminus S_t| \leq |S^\star| \leq B$ and the last equality follows from the greedy choice. Rearranging yields

$$f(S_{t+1}) - f(S_t) \geq \frac{1}{B}\big(f(S^\star) - f(S_t)\big).$$

Letting $g_t := f(S^\star) - f(S_t)$ gives $g_{t+1} \leq (1 - 1/B)g_t$, hence

$$f(S_B) \geq \big(1 - (1 - 1/B)^B\big)f(S^\star) \geq (1 - 1/e)\,f(S^\star),$$

as $(1 - 1/B)^B \leq e^{-1}$. This proof is standard; see, e.g., Nemhauser et al. (1978). $\qquad\square$

## B    Modeling Choices and Extensions

This section summarizes key modeling choices behind our framework and sketches extensions that we leave for future work. Throughout, we emphasize how these choices affect the interpretation of our indices and dynamics rather than proposing a single canonical model.

### B.1    Behavioral Types and Action Sets

For clarity, the main text adopts a minimal behavioral abstraction with two archetypal client types and simple action sets.

**Baseline types.**    We consider a population in which each client $i$ has a latent type $\tau_i \in \{\text{honest}, \text{gaming}\}$, with a fixed fraction of gaming types in simulations and the FL experiment. Honest types select actions from a constrained set

$$\mathcal{A}_i^{\text{align}} \subseteq \mathcal{A}_i,$$

which encode participation, local training, and reporting policies that aim to improve genuine welfare (e.g., standard local training on $D_i$ and truthful reporting of updates). Gaming types select from the full set $\mathcal{A}_i$, which additionally includes actions that target metrics or rewards while holding welfare flat or decreasing (e.g., discarding tail data, overfitting to public validation, or perturbing reports to influence leaderboards).

Within each type, the main experiments use simple stationary strategies: honest clients apply a fixed local training pipeline and reporting rule; gaming clients apply a fixed metric-targeting rule that is independent of history except through the current model $\theta_t$. This allows us to isolate how the design policy $\pi$ affects welfare, metrics, and participation even when behavioral complexity is limited.

**Extensions in behavioral richness.**    Several extensions are natural but beyond our scope:

- *Continuous heterogeneity:* instead of discrete types, clients could have continuous parameters (e.g., cost of effort, risk aversion, penalty sensitivity), with $\tau_i$ drawn from a distribution. Best responses and thresholds would then be functions of these parameters.

- *Adaptive and learning strategies:* clients could update their strategies over time using reinforcement learning or no-regret dynamics, learning how to trade off gaming and cooperation given observed rewards and sanctions.

- *Coalitions, Sybils, and collusion:* richer action sets could explicitly include coalition formation, Sybil identity creation, and coordinated reporting, rather than treating coalition effects only at the aggregate level in the dynamics layer.

Our indices and thresholds are defined at the level of strategy profiles and deviations, so they extend to these richer settings as long as welfare $W(\sigma)$ and metrics $M(\sigma)$ are well defined. The main trade-off is practical: more complex behavioral models make it harder to estimate the indices empirically and to connect them to concrete design levers.

## B.2 Alternative Welfare and Metric Definitions

The framework deliberately separates *welfare* from *metrics*, recognizing that real deployments may care about multiple objectives while exposing only a subset through observable scores.

**Scalar welfare and proxy metrics.** In the main text, welfare is modeled as a scalar functional

$$W(\sigma) = W(\theta; P^\star),$$

such as accuracy or utility on a target distribution $P^\star$ that encodes the deployment population and business or social objective. In the Fashion-MNIST experiment, we instantiate this as tail-class accuracy on classes 5–9, treating performance on these classes as the welfare outcome of interest.

Metrics $M(\sigma)$ are defined as proxy functionals constructed from evaluation pipelines, for example:

- head-class accuracy on a public validation split (used as a reward-driving head metric);

- overall test accuracy, which may be monitored but not explicitly rewarded;

- auxiliary diagnostics or fairness indicators, which may or may not enter incentives.

The Price of Gaming and Manipulability Index are defined abstractly in terms of $(W, M)$ and therefore apply to any such choices.

**Alternative scalarizations.** In settings with multiple objectives (e.g., subgroup performance, latency, cost), one can still fit into our scalar framework by using a scalarization of the form

$$W(\sigma) = U\big(W^{(1)}(\sigma), \ldots, W^{(L)}(\sigma)\big),$$

where $W^{(\ell)}$ are component-wise objectives and $U$ is a designer-chosen aggregator (e.g., weighted sum, minimum over groups, or a constrained utility that assigns $-\infty$ to infeasible fairness violations). Different scalarizations correspond to different normative choices and will in general induce different values of PoG and $\mathcal{M}(\pi)$ even under the same behavior and metrics.

Similarly, the exposed metric $M(\sigma)$ can be a vector, with only some coordinates entering rewards. Our formal definitions extend by either focusing on the metric components that are rewarded or by mapping $M$ to a scalar proxy $m(M)$ that captures how incentives are actually computed.

## B.3 Sketches for Multi-objective and Fairness-aware Welfare

We briefly sketch how the framework can be extended when welfare is explicitly multi-objective or fairness-aware.

**Vector-valued welfare.** Suppose welfare is a vector

$$\mathbf{W}(\sigma) = \big(W^{(g)}(\sigma)\big)_{g \in \mathcal{G}},$$

where $g$ indexes groups, clients, or objectives (e.g., accuracy by demographic group, latency, and cost). A simple extension is to define group-specific Prices of Gaming

$$\mathrm{PoG}^{(g)}(\pi) = \frac{W^{(g)}(\sigma^{\mathrm{align}}) - W^{(g)}(\sigma^{\mathrm{game}})}{W^{(g)}(\sigma^{\mathrm{align}})},$$

and to monitor both the worst-case and average group PoG. This distinguishes regimes where gaming primarily harms particular groups from those where losses are more evenly spread.

**Fairness-aware aggregations.** Alternatively, one can define a fairness-aware scalar welfare, for example:

- *Min-based:* $W(\sigma) = \min_{g \in \mathcal{G}} W^{(g)}(\sigma)$, emphasizing the worst-off group.

- *Penalty-based:* $W(\sigma) = \bar{W}(\sigma) - \lambda \cdot \mathrm{Disp}(\sigma)$, where $\bar{W}$ is average welfare, Disp is a disparity measure (e.g., gap between best and worst group), and $\lambda \geq 0$ trades off performance and fairness.

- *Constraint-based:* $W(\sigma)$ is defined only over profiles satisfying fairness constraints (e.g., equalized error rates), with infeasible profiles treated as having very low or undefined welfare.

Our indices then quantify how gaming and cooperation affect both overall performance and fairness, depending on the chosen $W$.

**Implications for design.** Multi-objective and fairness-aware welfare primarily affect:

- how designers define aligned benchmarks $\sigma^{\mathrm{align}}$ (e.g., fairness-satisfying equilibria);

- which components of $\mathbf{W}(\sigma)$ are reflected in metrics and rewards;

- how Prices of Gaming and Cooperation are interpreted across groups.

The structural role of the indices and dynamics remains unchanged: they still measure how far metric-targeting behavior can drift from the chosen welfare definition and how participation responds. A systematic treatment of fairness-aware incentives in federated settings is an important direction for future work and would likely require combining our framework with group-specific constraints and fairness-sensitive audit mechanisms.

## C   Simulation Setup and Hyperparameters

This appendix summarizes the main modeling and hyperparameter choices for the stylized simulator and for the penalty and information-design sweeps reported in Section 7. The goal is to make the experiments reproducible at a high level without tying the framework to a specific implementation.

### C.1   Stylized Simulator

**Environment and population.** The stylized simulator instantiates the strategic FL model from Section 3 in a simplified cross-silo setting with a fixed population of $n$ clients. Each client is typed as either *honest* or *gaming*, with a fixed fraction of gaming types (set to 30% in the baseline experiments). Types remain fixed over time. All clients hold local datasets drawn from heterogeneous but stationary distributions $\{\mathcal{P}_i\}_{i \in \mathcal{I}}$, and the welfare distribution $P^\star$ is a mixture of these $\mathcal{P}_i$.

At each round $t$, the server maintains a global model $\theta_t$ and broadcasts it to all eligible clients. Participating clients apply a local training rule (e.g., a fixed number of SGD steps on $D_i$) to produce an internal update $u_{i,t}^{\mathrm{int}}$, which is then transformed into a reported update $u_{i,t}$ according to the client's type and strategy. The server aggregates reported updates via a fixed aggregation rule (a FedAvg-style weighted mean in our implementation) and evaluates the updated model using a held-out evaluation pipeline.

**Behavioral types and actions.** Honest clients select actions from $\mathcal{A}_i^{\mathrm{align}}$, which in the simulator are implemented as:

- participation whenever expected utility is above a client-specific threshold;

- standard local training on $D_i$ without discarding examples;

- truthful reporting $u_{i,t} = u_{i,t}^{\mathrm{int}}$ (up to any mandated privacy perturbation).

Gaming clients select from $\mathcal{A}_i$, which extends $\mathcal{A}_i^{\text{align}}$ with simple metric-targeting behaviors. Concretely, the gaming strategy used in the main experiments:

- emphasizes subsets of data that are overrepresented in the public evaluation (e.g., "head" groups);

- downweights or discards data that primarily contributes to welfare but has little effect on the public metric;

- optionally perturbs updates in directions that improve the disclosed metric while leaving welfare flat or reduced.

Both types use myopic best responses with respect to the current design policy $\pi$ and their outside option, as in Section 5.

**Welfare, metrics, and participation.** For the stylized experiments, welfare $W_t$ is instantiated as expected performance on the welfare distribution $P^\star$, normalized to $[0,1]$. The metric $M_t$ is a scalar proxy constructed from the same family of losses but evaluated on a distinct metric distribution and with a different weighting over clients and groups; this distribution is chosen so that gaming behaviors can move $M_t$ without proportionate changes in $W_t$. The aggregate participation rate is

$$x_t = \frac{1}{n} \sum_{i \in \mathcal{I}} \mathbb{I}\{p_{i,t} = 1\},$$

and we report steady-state averages

$$\overline{W}, \quad \overline{M}, \quad \overline{x}$$

computed over post–burn-in rounds. The Price of Gaming is computed by comparing $\overline{W}$ under aligned and mixed-type configurations, as described in Section 4.3.

**Hyperparameters and averaging.** Each simulation run proceeds for a fixed number of rounds with an initial burn-in period discarded to reduce transient effects. Unless otherwise specified, we:

- fix the fraction of gaming clients, the aggregation rule, and the local training pipeline across runs;

- vary only the design levers under study (e.g., penalty strength or public-metric weight);

- average reported quantities over multiple random seeds (affecting client initialization, data sampling, and evaluation noise).

The exact numerical values of $n$, the number of rounds, and the learning-rate schedule are not critical for interpreting our indices, and were chosen to balance stability with computational cost.

## C.2 Penalty and Information-design Sweeps

**Penalty-strength sweep.** The penalty-strength experiments in Table 2 vary a scalar sanction parameter $\alpha_{\text{penalty}}$ while keeping all other components of the design policy $\pi$ fixed. For each value in a predefined grid (e.g., $\alpha_{\text{penalty}} \in \{0.3, 0.5, 0.7, 1.0, 1.5\}$):

- we instantiate a policy $\pi(\alpha_{\text{penalty}})$ that scales expected sanctions linearly in a violation score, as in Assumption 4.11;

- run the simulator to steady state with a fixed fraction of gaming clients;

- record $\overline{W}_{\text{game}}$, $\overline{x}_{\text{game}}$, and the resulting PoG relative to the aligned benchmark.

This allows us to trace how increasing penalty strength moves the system toward or beyond the minimal sanction level $\alpha_{\text{min}}$ and the benign boundary $\alpha_{\text{benign}}$ introduced in Section 4.5.

**Information-design sweep.** The information-design experiments in Table 3 vary the public-metric weight $\rho_{\text{pub}}$ in a mixed challenge policy (Section 6.2) while holding other levers fixed. For each $\rho_{\text{pub}} \in \{1.0, 0.8, 0.6, 0.4, 0.2\}$:

- the overall metric is defined as

$$M_t = \rho_{\text{pub}} \, M_t^{\text{pub}} + (1 - \rho_{\text{pub}}) \, M_t^{\text{priv}},$$

  where $M_t^{\text{pub}}$ is fully disclosed and $M_t^{\text{priv}}$ is based on private or randomized challenges;

- reward rules depend on $M_t$ through a fixed monotone function, so changing $\rho_{\text{pub}}$ alters the relative importance of public and private signals without changing the reward shape;

- audits and sanctions are held constant so that observed changes in $\overline{W}_{\text{game}}$, $\overline{x}_{\text{game}}$, $\overline{M}_{\text{game}}$, and PoG can be attributed to information design alone.

**Reporting and robustness.** For both sweeps, we report steady-state averages over post–burn-in rounds and aggregate results across seeds. Individual runs exhibit stochastic variation, but the qualitative patterns in Tables 2 and 3—improved welfare and lower PoG in a benign penalty band, and narrower metric–welfare gaps but potentially higher PoG under aggressive downweighting of public metrics—are stable across reasonable choices of simulator hyperparameters.

## D  Fashion-MNIST Federated Experiment Details

This appendix summarizes the setup of the Fashion-MNIST experiment in Section 7, including the partitioning scheme, client types, training protocol, and a few brief robustness checks. The goal is to provide enough detail to reproduce the qualitative patterns in Table 4 without tying the framework to a particular implementation.

### D.1  Partitioning, Heterogeneity, and Client Types

**Dataset and head/tail split.** We use the standard Fashion-MNIST dataset with 60,000 training and 10,000 test examples across ten classes labeled 0–9. For the experiment, we designate classes 0–4 as *head* classes and classes 5–9 as *tail* classes. Welfare is defined as accuracy on the tail classes, evaluated on a held-out test split, while the public head metric is accuracy on the head classes, evaluated on a public validation split (Section 7).

**Client partitioning and heterogeneity.** The training portion of Fashion-MNIST is partitioned across $n = 30$ clients. To keep the focus on strategic behavior rather than extreme data skew, we use a mild label-heterogeneous partition:

- each class is first shuffled and split into 30 shards of approximately equal size;

- for each client $i$, we sample a small number of shards per class so that all clients observe both head and tail classes, but with modest variation in proportions;

- this yields client datasets $D_i$ with overlapping but non-identical label distributions, avoiding degenerate clients that only see head or only tail labels before any gaming behavior.

The public validation split is constructed analogously from head-class examples only; the tail-class test split is kept hidden from clients and used solely for welfare evaluation on the server side.

**Client types and gaming behavior.** We consider two fixed client types:

- *Honest clients* follow a standard local training procedure on their full local dataset $D_i$ and report their updates truthfully (up to any noise added by the protocol).

- *Gaming clients* follow a head-focused strategy: before local training, they discard or heavily down-weight all examples from tail classes 5–9, train only on head-class data, and implicitly optimize toward performance on the public head-only validation split. They do not inject arbitrary model poisoning and remain consistent over rounds.

In the aligned scenario, all 30 clients are honest. In the gaming scenario, 30% of clients (randomly chosen at initialization and fixed thereafter) are gaming clients. Client types are not observable to the server; they are inferred only through their effect on metrics and welfare.

## D.2 Training Protocol and Hyperparameters

**FL protocol.** We use a standard cross-silo FedAvg-style protocol:

- global rounds: $T = 40$;

- all 30 clients participate in every round (no client sampling);

- the server maintains a single global model $\theta_t$ and aggregates client updates via a weighted average proportional to local data size.

We run the protocol twice, once with all clients honest and once with the mixed population described above, using identical random seeds and hyperparameters except for client behavior.

**Model and optimization.** The global model is a small convolutional network suitable for Fashion-MNIST, with two convolutional layers followed by a fully connected head and a softmax output. We use cross-entropy loss for local training and a standard optimizer (e.g., SGD or Adam) with:

- a fixed learning rate over rounds;

- mini-batch training with a moderate batch size;

- a small, fixed number of local epochs per round for each client.

Exact layer widths, learning rates, and batch sizes are chosen so that the aligned configuration reaches a test accuracy around 0.88 on the full test set, but otherwise follow standard Fashion-MNIST baselines. They are not critical for the qualitative comparisons reported in Table 4.

**Evaluation and metrics.** At the end of each round, the server evaluates the current global model on:

- a public validation split restricted to head classes 0–4, yielding the head metric $M_{\text{head},t}$;

- a hidden test split restricted to tail classes 5–9, yielding tail welfare $W_{\text{tail},t}$;

- an optional full test split across all ten classes, yielding overall accuracy $A_{\text{full},t}$.

The values $\overline{M}_{\text{head}}$, $\overline{W}_{\text{tail}}$, and $\overline{A}_{\text{full}}$ reported in Table 4 are averages over the last ten rounds. The Price of Gaming in this experiment is computed as

$$\text{PoG} \approx \frac{\overline{W}_{\text{tail}}^{\text{aligned}} - \overline{W}_{\text{tail}}^{\text{gaming}}}{\overline{W}_{\text{tail}}^{\text{aligned}}}.$$

### D.3 Additional Robustness Checks (Brief)

To check that Table 4 is not an artifact of a single configuration, we performed a small set of robustness checks:

- **Varying the fraction of gaming clients.** We repeated the experiment with 20% and 40% gaming clients. As expected, the metric–welfare gap and PoG increased with a higher gaming fraction and decreased when fewer clients gamed, while the qualitative pattern (improved head metric, degraded tail welfare) remained.

- **Alternative label partitions.** We swapped the head and tail roles of certain classes (e.g., using a different subset of five classes as tail) and observed similar behavior: when clients strategically focus on the classes emphasized by the public metric, performance on de-emphasized classes degrades even if overall accuracy changes little.

- **Random seeds and mild hyperparameter changes.** Across multiple random seeds and modest variations in local learning rate and number of local epochs, the aligned configuration consistently outperformed the gaming configuration on tail welfare, while the gaming configuration maintained a higher head-only public metric.

These checks are not meant to be exhaustive, but they support the claim that the Fashion-MNIST experiment illustrates a robust instance of the high-metric, low-welfare pattern predicted by our framework, rather than a fragile consequence of a particular training run.

