# OpenReview forum: "Gaming and Cooperation in Federated Learning: What Can Happen and How to Monitor It"
_TMLR — Accepted by TMLR_

### Review · Reviewer_UD5M · 2026-01-01

**Summary Of Contributions:**

**Summary:**

This paper frames federated learning (FL) as a governed strategic system rather than a static optimization problem, and proposes a compact, game-theoretic language to reason about metric gaming, cooperation, participation dynamics, and governance levers. The core contributions are indices that quantify manipulability (M), the price of gaming (PoG), and the price of cooperation (PoC), along with a resilience indicator for participation dynamics and critical sanction thresholds that separate under-enforcement from over-deterrence. The authors claim threshold conditions, an audit-budget allocation rule with a (1−1/e) guarantee, early-warning indicators, and a design toolkit; stylized simulations and a Fashion-MNIST case study are said to qualitatively and quantitatively validate the framework.

**Strengths:**
- The paper addresses an important and timely gap: the interplay between privacy, limited observability, incentives, and cooperation/collusion in FL deployments.
- The authors recast FL as a governed strategic system and introduce a metric-based lens (M, PoG, PoC) that explicitly separates metric performance from welfare, connecting Goodhart-style failures to incentives in FL.
- They derive interpretable thresholds (α_min, α_benign) and a resilience indicator R(π) that link evaluation/audit design to participation stability and tipping points.
- They propose a unifying Eval-Info-Reward-Audit architecture that can place diverse existing mechanisms under one umbrella and reason about incentive effects beyond aggregation rules.


**Weaknesses:**
- The indices (especially M(π) and PoG(π)) rely on suprema over strategy sets and equilibria that are hard to characterize; the paper’s practical estimation procedures are heuristic and may be sensitive to modeling choices.
- Several key claims (e.g., the (1−1/e) audit allocation guarantee) are only mentioned at a high level in the excerpt; formal problem statements, assumptions, and proofs are necessary to assess rigor.
- Empirical illustrations appear limited to stylized simulations and a Fashion-MNIST case study, which may be insufficient to support strong claims about generality and operational guidance for diverse, large-scale FL deployments.
- It is unclear how the proposed indices behave under adaptive, multi-round poisoning strategies and advanced defenses prevalent in recent literature; no head-to-head evaluation against state-of-the-art attacks/defenses is described.
- The definition of welfare $W$ and metrics $M$ in concrete FL contexts (e.g., with privacy, distribution shift, and fairness constraints) could be better operationalized to aid adoption.
- While the related work section is broad and well-situated, explicit empirical comparisons to recent robust aggregation and poisoning frameworks (e.g., PoisonedFL, FedCC, AdaAggRL) are absent; such comparisons would better position the proposed indices and toolkit in practice.

**Audience:**

Yes

**Audience Explanation:**

The paper addresses an important and timely gap: the interplay between privacy, limited observability, incentives, and cooperation/collusion in FL deployments. The work offers technical novelty by recasting Federated Learning (FL) as a governed strategic system rather than a static optimization problem. This reframing allows for reasoning about incentive effects beyond simple aggregation rules.

**Broader Impact Concerns:**

The primary concern is that miscalibrated governance thresholds or overzealous enforcement could unintentionally deter benign cooperation and degrade system welfare. Consequently, the work requires more robust guidance on uncertainty-aware calibration to mitigate the risk of suppressing legitimate participation.

**Claims And Evidence:**

No

**Claims Explanation:**

While the paper presents substantial empirical results, several claims lack adequate support:
- **Missing Theoretical Evidence:** Key theoretical claims, such as the audit-budget allocation rule with a $(1-1/e)$ guarantee, are only mentioned at a high level.
- **Limited Empirical Scope:** The experimental validation is restricted to "stylized simulations and a Fashion-MNIST case study".
- **Lack of Comparative Validation:** The paper lacks head-to-head evaluations against state-of-the-art adaptive attacks (e.g., PoisonedFL) or modern defenses (e.g., FedCC, AdaAggRL). Consequently, it is unclear if the proposed indices actually track true risk under realistic threat models.

**Requested Changes:**

Refer to the weaknesses above.

---

> ### Author Response · Authors · 2026-01-26
> **Response to Reviewer UD5M (Rigor, Estimation Reliability, Privacy/Noise, and Modern Attack–Defense Evaluation)**
>
> ### Dear Reviewer UD5M,
>
> Thank you for the careful and constructive review. We appreciate your recognition of the paper’s motivation and the value of separating server-visible metrics from operational welfare. In this revision, we focused on strengthening the rigor and empirical grounding in the exact areas you highlighted.
>
> ### Quick navigation (where to find the updates)
>
> * Formal theory and guarantees: Section 6.3 (main text) and Appendix A.3 (full details).
> * Additional empirical studies requested by the review: new paragraphs appended after the existing results in Section 7.2, in the following order:
>
>   * Estimator reliability under partial audits
>   * Noise (privacy) and auditability: persistent metric–welfare separation
>   * Modern attack–defense replication: metric–welfare separation under contemporary threats
>
> ### Theory: clearer problem statements, assumptions, and proofs (including the audit allocation guarantee)
>
> You noted that several key claims were previously presented at a high level, making it difficult to assess rigor. We addressed this by rewriting Section 6.3 to include explicit problem statements, assumptions, and theorem/proposition-level claims, and by placing the detailed derivations and proofs in Appendix A.3. The revised presentation makes transparent what is being optimized, what class of strategic behaviors is considered, and what conditions are required for the stated approximation guarantee.
>
> ### Practical estimation: reliability of audit/log-based estimation under partial audits
>
> You raised a concern that the indices may be difficult to estimate in realistic deployments and that retrospective/log-based estimation could be brittle. In response, we added the new Section 7.2 paragraph titled "Estimator reliability under partial audits." This study quantifies how reliably a budget-limited auditor can recover welfare loss signals when only a fraction of clients is audited, reporting rank consistency, estimation variability, and error behavior around governance-relevant thresholds. The goal is to make estimation reliability measurable rather than assumed.
>
> ### Privacy/noise and auditability: controlled noise injection and its effect on governance signals
>
> You suggested grounding the privacy–auditability–incentive interaction with an explicit experiment that injects noise into updates or audit signals. We added the new Section 7.2 paragraph titled "Noise (privacy) and auditability: persistent metric–welfare separation," where we introduce clipping and Gaussian noise (noise-multiplier sweep) and compare aligned vs. gaming profiles under the same noise level. The results show how privacy-driven noise can reduce audit resolution and sustain, and in some regimes amplify, the gap between server-visible metrics and tail welfare.
>
> ### Modern comparative validation: adaptive multi-round attacks and contemporary defenses
>
> You pointed out that it was unclear how the framework behaves under modern adaptive poisoning strategies and stronger contemporary defenses, and that head-to-head validation was missing. We added the new Section 7.2 paragraph titled "Modern attack–defense replication: metric–welfare separation under contemporary threats," which evaluates adaptive multi-round model poisoning and backdoor/model-replacement style attacks alongside representative modern defenses, and reports both the public metric and operational welfare. This is intended to place the framework in threat/defense settings closer to recent practice.
>
> Thank you again for the review. Your comments directly shaped these revisions, and we hope the updated Section 6.3 / Appendix A.3 and the appended studies in Section 7.2 address your concerns about rigor, estimator reliability, privacy/noise grounding, and modern comparative validation.
>
> P.S. We also made minor manuscript-wide polish changes (typos/cross-references, clearer Discussion organization including the added experimental notes, and converting some bar plots to tables with exact values) to improve clarity and reproducibility.
>
> Sincerely,
>
> The Authors

---

### Review · Reviewer_edNK · 2026-01-17

**Summary Of Contributions:**

This paper shifts the perspective of Federated Learning (FL) from a standard distributed optimization problem to a governed strategic system. It introduces an analytical framework designed to distinguish between genuine welfare-improving contributions and "metric gaming," where participants manipulate reported updates to maximize rewards without improving actual performance. It introduces three novel indices to measure the health of an FL system: the Manipulability Index (capacity for gaming), the Price of Gaming (welfare loss due to manipulation), and the Price of Cooperation (distinguishing benign collaboration from harmful collusion). The theories are validated through both stylized simulations and a practical Fashion-MNIST experiment demonstrating how high-metric/low-welfare regimes emerge in real-world training.

Strength:
* The paper is clearly written and easy to follow.
* The literature review is thorough and well-organized.
* The proposed framework provides a unified lens for analyzing diverse FL incentive designs under a strategic/governed-system perspective, and offers useful guidance for reducing manipulability and improving federation stability.

Weakness:
* While the proposed indices are theoretically well-motivated, they may be difficult to compute or estimate in realistic FL deployments. The authors suggest retrospective/log-based estimation as a practical workaround, but the reliability and robustness of these approximations are not sufficiently demonstrated.
* The Fashion-MNIST experiment provides supportive evidence, but the setup appears somewhat unrealistic. In real-world FL systems, the gap between “metric” and “welfare” may be less pronounced. It is therefore unclear whether the proposed approximate indices remain informative and stable when this gap is small or noisy.
* The paper argues that stronger privacy mechanisms (e.g., Differential Privacy) may reduce the resolution of audit signals and thus increase manipulability. However, the experiments do not explicitly model this privacy–auditability–incentive trade-off. Even a simple experiment that injects noise into updates or audit signals would strengthen this claim and provide empirical grounding.

**Audience:**

Yes

**Audience Explanation:**

The paper offers a useful and general perspective on incentive design in federated learning, which complements prior work that often focuses on specific incentive mechanisms or threat models. By formalizing manipulation risks and proposing interpretable indices, the framework can help researchers and practitioners reason about stability and strategic behavior in FL systems.

**Broader Impact Concerns:**

No concerns.

**Claims And Evidence:**

Yes

**Claims Explanation:**

The main ideas and theoretical development are clearly presented, and the experiments provide reasonable support for the core claims. Overall, the evidence is consistent with the paper’s conclusions, though additional empirical validation would strengthen the practical relevance of the proposed indices

**Requested Changes:**

I suggest that the authors to address the weakness, such as:

* Provide a controlled experiment simulation where the ground-truth indices are known, then compare them against the estimates produced by the proposed retrospective/log-based method. This would clarify the reliability and "false-negative" rate of using logs for governance.

* Evaluate the framework in a setting where the evaluation metric is a closer proxy for true welfare (i.e., higher correlation than the Head/Tail split used in Table 2). This will demonstrate whether the indices remain informative when gaming is subtle rather than overt.

* Provide an experiment that incorporates controlled noise injection (e.g., Differential Privacy) into client updates. This supports the theoretical claim that privacy-enhancing technologies increase the Price of Gaming by reducing audit signals.

---

> ### Author Response · Authors · 2026-01-26
> **Response to Reviewer edNK (Estimation Reliability, Privacy/Noise Trade-offs, and High-Alignment Metrics)**
>
> ### Dear Reviewer edNK,
>
>
> Thank you for the thoughtful review and for pointing out where the empirical validation and operational framing could be strengthened. In this revision, we focused on the specific issues you raised around the reliability of audit-based estimation, the interaction between privacy/noise and auditability, and whether our claims persist when the server-visible metric is more closely aligned with operational welfare.
>
> ### Quick navigation (where to find the updates)
>
> All new empirical studies addressing your comments are provided as new paragraphs appended after the existing results in Section 7.2, in the following order:
>
> * Estimator reliability under partial audits
> * Noise (privacy) and auditability: persistent metric–welfare separation
> * High-alignment metrics: diminishing but persistent metric–welfare separation
>
> ### Estimation reliability: audit/log-based estimation under partial audits
>
> You raised concerns about whether audit-based or retrospective estimation is reliable when only limited audit capacity is available, and whether governance decisions could be sensitive to estimation error. To address this directly, we added the Section 7.2 paragraph titled "Estimator reliability under partial audits." This study evaluates how well a budget-limited auditor can recover welfare-loss signals when only a fraction of clients is audited. We report rank consistency, estimation variability, and error behavior around governance-relevant thresholds, clarifying when the estimator is dependable and how uncertainty decreases as audit coverage increases.
>
> ### Privacy/noise and auditability: explicit stress test under noisy updates
>
> You emphasized that privacy mechanisms or noise injection can reduce observability and therefore affect both auditability and incentives. We added the Section 7.2 paragraph titled "Noise (privacy) and auditability: persistent metric–welfare separation." In this experiment, client updates are clipped and perturbed with Gaussian noise (noise-multiplier sweep), and we compare aligned versus gaming profiles at the same noise level. The results quantify how reduced audit resolution interacts with strategic behavior, and show that the separation between the server-visible metric and tail welfare persists under privacy/noise, with clear implications for governance calibration.
>
> ### More realistic alignment: when the public metric is closer to welfare
>
> You also noted that the gap between observable metrics and welfare can be smaller in practice, and asked whether our conclusions still hold when the server-visible score is more aligned with what matters operationally. To test this, we added the Section 7.2 paragraph titled "High-alignment metrics: diminishing but persistent metric–welfare separation." We define a mixed public metric that interpolates between a purely head-based metric and a more welfare-aligned metric, and evaluate both aligned and gaming profiles across alignment levels. The results show that increasing alignment reduces the incentive and magnitude of manipulation, but does not eliminate the separation entirely, which supports the paper’s broader claim that governance risk can remain even under better-aligned scoring.
>
> Thank you again for the review. We hope these additions in Section 7.2 address your concerns about estimator reliability, privacy/noise trade-offs, and the robustness of our conclusions under more aligned public metrics.
>
> P.S. We also made minor manuscript-wide polish changes (typos/cross-references, clearer Discussion organization including the added experimental notes, and converting some bar plots to tables with exact values) to improve clarity and reproducibility.
>
> Sincerely,
>
> The Authors

---

### Review · Reviewer_AFn6 · 2026-01-20

**Summary Of Contributions:**

This paper frames federated learning (FL) not merely as a distributed optimization problem, but as a strategic principal-agent system in which clients may cooperatively improve welfare or strategically “game” metrics. The authors propose a **three-layer analytical framework** for understanding (and mitigating) such behavior, and considered mechanism and information design for each of the layers:

1. Layer 1: Metric layer. The authors define manipulability, Price of Gaming (PoG) and Price of Cooperation (PoC), reflecting how misaligned metrics could be gamed to harm social welfare, and cooperation could either help or harm welfare. Threshold arguments are also given to establish when sanctions block harmful gaming without suppressing beneficial cooperation.
2. Layer 2: Dynamics layer. The authors model the participation of clients as a dynamic process that could be unstable. The authors define tipping point and domino exit, and provide condition for resilience. Coalition effects and warning signals are also studied, and auto-switch rules are proposed to prevent participation collapse.
3. Layer 3: Design toolkit layer. The authors then propose a set of mechanism design tools through levers of different component that steer different metrics to favorable directions. Governance checklists and audit budget allocation with approximate guarantee are given.

At last, this framework is validated through simulations and a federated Fashion-MNIST case study, showing divergences between welfare and observed metrics, and illustrating how audit & evaluation design can mitigate gaming.

**Audience:**

Yes

**Audience Explanation:**

This paper presents a model on the strategic model of federated learning. This is a realistic and interesting problem for machine learning researchers.

**Broader Impact Concerns:**

The framework highlights vulnerabilities in federated learning to strategic manipulation and collusion, which could be exploited in real deployments if detection and governance mechanisms lag behind.

**Claims And Evidence:**

No

**Claims Explanation:**

I have the following concerns regarding the presentation and soundness of the paper.

1. Many notions in the paper are mentioned but never formally defined. For example, what is the definition of "individually rational", and what is "coalition-rational" in Section 4.5? What is the "contraction" used in the proof sketch of Proposition 5.9? What is a "lever" in Section 6? What is a "reference class" in the statement of Proposition 6.2? These terminologies may be standard in economics and mechanism design, but I don't think they are necessarily within the context of machine learning community.

2. The model seems new but links from the model towards existing models in federated learning/mechanism design are not discussed. Section 3 presents a model that models federated learning as a strategic system. However, the connection between this model and the models studied by previous works are not discussed. Is it a generalization of previously studied models, or a completely new concept? How does it fit into the specific real-world scenarios of federated learning?

3. Some proofs can be stated more formally and rigorously. For example, only a proof sketch is provided for Proposition 4.13. It seems to me that Proposition 5.4 cannot hold because Assumption 5.2 doesn't give any structure on the dynamics of $x_t$ (or Assumption 5.2 is falsely stated, I assume it should be "client $i$ participates at round $t+1$ iff..." ).

**Requested Changes:**

1. Formalize the statement and proofs of the theorems.

2. Discuss the similarities and differences between the proposed model and models studied by other works.

3. The theoretical part of the paper doesn't seem to be related to federated learning. Please specify what aspects of federated learning is required for the theorems to hold (otherwise the same result would hold for a general mechanism design/game theoretic principal-agent framework, and it is unclear how the results apply to federated learning).

4. The theoretical results seem quite straightforward, and the scope of the theoretical results are way more specific than the model itself. Please specify the following: What are the technical challenges on proving the theorems? How to justify that the assumptions are realistic in federated learning?

---

> ### Author Response · Authors · 2026-01-26
> **Response to Reviewer AFn6 (Definitions, Model Positioning, Proof Formalization, and FL-Specificity)**
>
> ### Dear Reviewer AFn6,
>
> Thank you for the careful reading and for clearly identifying places where the presentation and soundness needed improvement. We agree that the previous version left some terms under-defined, did not sufficiently position the model relative to prior FL and mechanism-design views, and relied too heavily on proof sketches. In this revision, we focused on making the theory precise, checkable, and clearly tied to FL protocol components.
>
> ### Quick navigation (where to find the updates)
>
> * Definitions and terminology (IR, coalition-rationality, “lever,” “reference class”): Section 4.5 and the beginning of Section 6; notation cleanup in Section 3.
> * Participation dynamics assumptions and participation map: Section 5.1.
> * Strengthened formal proofs (including the contraction-based argument): Section 5.2 and the corresponding appendix proof material.
> * “Where FL enters” + relation to prior work: Section 3 and short clarifying paragraphs around key propositions in Sections 4–6.
> * Broader impact / dual-use framing: Broader Impact Statement.
>
> ### Undefined terminology and notation
>
> You noted several notions were used without formal definition. We now define them at first use and align notation so later sections do not depend on implicit meanings. Section 4.5 formalizes individual rationality and coalition-rationality via baseline reference behavior and welfare comparisons. The dynamics proof material now states the contraction condition explicitly and specifies the map and conditions under which it is contractive. At the beginning of Section 6, we define a design “lever” as an adjustable policy coordinate and a “reference class” as a deployment-relevant subset of behaviors/strategies. Section 3 now introduces the ambient strategy space so the subset relation is fully grounded.
>
> ### Model positioning and FL grounding
>
> We strengthened Section 3 to clarify how the framework relates to principal–agent / mechanism-design perspectives, and what is specific to FL. In particular, protocol-level choices (evaluation, information disclosure, reward, audit) act as governance objects, and incentives are mediated through the evaluation and information-release pipeline rather than direct access to welfare. This is intended as a governance lens tied to concrete FL protocol components, not a generic game-theory abstraction with FL terminology.
>
> ### Proof rigor and assumption–proposition consistency
>
> We replaced or strengthened key proof sketches with checkable arguments. For the participation-dynamics result relying on contraction, we now state the needed conditions and provide a standard fixed-point argument, with full proof details in the appendix. We also rewrote Section 5.1 to “lock” the participation rule so the participation condition, induced participation map, and downstream propositions are consistent (including the inequality direction and its interpretation), removing ambiguity in how next-round participation is derived from utility comparisons and the type distribution. Threshold-style results were also strengthened by making assumptions explicit and clarifying why the stated ordering/existence follows under those conditions.
>
> ### Technical challenges and realism
>
> We now make explicit that while some mathematical steps are standard, the core challenge is operational: designing evaluation, disclosure, and auditing under partial observability, noise, and strategic adaptation so that the assumptions are plausibly satisfied and governance signals remain informative.
>
> ### Broader impact / dual-use
>
> We revised the Broader Impact Statement to acknowledge dual-use risk and emphasize uncertainty-aware calibration and safeguards to reduce harm from over-enforcement or miscalibrated thresholds, while avoiding operational details that would directly facilitate exploitation.
>
> Thank you again for the detailed critique. We hope the revised manuscript addresses your concerns about definitions, positioning, rigor, and FL-specific grounding.
>
> P.S. We also made minor manuscript-wide polish changes (typos/cross-references, clearer Discussion organization including the added experimental notes, and converting some bar plots to tables with exact values) to improve clarity and reproducibility.
>
> Sincerely,
>
> The Authors

---

### Decision · Action_Editor_3vag · 2026-02-27

**Recommendation:** Accept with minor revision

**Additional Comments:**

For proposition 5.4, the reviewer pointed out that the relationship of $x_{t+1}$ to what happens at time $t$ is not defined.  The authors describe a number of changes meant to address this, but the reviewer and I both believe this issue remains.

Essentially, the issue as I see it is that the existing equations define $p_{i,t}$ as a (thresholded) best response to $x_t$ and then define $x_t$ in terms of $p_{i,t}$  That is, as written, $x_t$ must be a fixed point of these definitions.  $x_{t+1}$ first appears in equation (5.3) without definition, and looking at the proof does not give me clarity about how it is meant to be designed.

I see two possible resolutions.  One is that the current fixpoint version in intended, and so what is needed is an explicit definition of the dynamics by which $x_{t+1}$ is defined.  The other is that either $x_t$ or $p_{i,t}$ is actually meant to depend on the values at $t-1$, meaning it is not guaranteed to satisfy a fixpoint and instead defines dynamics.

Whichever version matches the authors' intention, the needed text changes are small, and therefore I have recommended a minor revision as a major revision with an additional round of review would be a waste of time and resources for an otherwise acceptable paper.  However, I cannot accept the final camera ready version until this issue is resolved.  Given that the issue has not been successfully addressed in the original revision this does mean there is a small risk I will not ultimately be able to accept the paper.

**Audience:**

Yes

**Audience Explanation:**

The reviewers unanimously agree that the paper is of interest to at least some individuals interested in federated learning.

**Claims And Evidence:**

Yes

**Claims Explanation:**

The reviewers unanimously agree that this standard is generally met.  There are some lingering concerns about the applicability of the model and the strength of the results, but nothing that rises to the point of being an issue for TMLR.

There is however one remaining issue regarding Proposition 5.4 where I agree with the reviewer that it does not appear to have been successfully addressed.  This needs resolution before final acceptance, as I discuss below.